# VERIFYBENCH: BENCHMARKING REFERENCE-BASED REWARD SYSTEMS FOR LARGE LANGUAGE MODELS

**Yuchen Yan**[1,2][*] **Jin Jiang**[2,3]**, Zhenbang Ren**[1,4]**, Yijun Li**[1]**, Xudong Cai**[1]**, Yang Liu**[2]**,**
**Xin Xu**[5]**, Mengdi Zhang**[2]**, Jian Shao**[1][†] **Yongliang Shen**[1][†]**, Jun Xiao**[1]**, Yueting Zhuang**[1]
[1]Zhejiang University   [2]Meituan Group   [3]Peking University
[4]University of Electronic Science and Technology of China
[5]The Hong Kong University of Science and Technology
{yanyuchen, syl, jshao}@zju.edu.cn

## ABSTRACT

Large reasoning models such as OpenAI o1 and DeepSeek-R1 have demonstrated remarkable performance in complex reasoning tasks. A critical component of their training is the incorporation of reference-based reward systems within reinforcement learning (RL), where model outputs are evaluated against ground truth references. However, existing reward benchmarks focus on preference comparisons between responses rather than evaluating verification against ground truth references, leaving a critical gap in our ability to evaluate verification systems used in reasoning model training. In this paper, we introduce VerifyBench and its challenging variant VerifyBench-Hard, two benchmarks specifically designed to assess reference-based reward systems. These benchmarks are constructed through meticulous data collection and curation, followed by careful human annotation to ensure high quality. Our comprehensive evaluation reveals that while larger model-based verifiers show promise on standard cases, all current systems demonstrate substantial room for improvement on challenging instances. Through systematic analysis of performance patterns across reasoning tasks and error categories, we provide insights for advancing reference-based reward systems. These benchmarks establish a standardized framework for improving verification accuracy, ultimately enhancing reasoning capabilities in models trained via RL.

|  | Project Page | https://zju-real.github.io/VerifyBench |
|---|---|---|
|  | Benchmark | https://huggingface.co/datasets/ZJU-REAL/VerifyBench |
|  | Code | https://github.com/ZJU-REAL/VerifyBench |

## 1 INTRODUCTION

In recent years, large language models (LLMs) have exhibited remarkable capabilities, significantly assisting humans across diverse practical domains (DeepSeek-AI et al., 2025b; Grattafiori et al., 2024; Yang et al., 2025). Reinforcement learning from human feedback (RLHF) has been crucial to this progress, with reward models playing a central role by evaluating and scoring model-generated responses to guide training. This approach has led to the development of numerous benchmarks (Lambert et al., 2025; Liu et al., 2024; Zhou et al., 2024) for systematic reward model evaluation, focusing primarily on pairwise preference judgments between competing responses.

The emergence of specialized large reasoning models (LRMs) (DeepSeek-AI et al., 2025a; Qwen Team, 2024; Kimi Team et al., 2025) such as OpenAI's o1(OpenAI, 2024) and DeepSeek-R1(DeepSeek-AI et al., 2025a) has fundamentally changed this landscape. These models achieve unprecedented performance on reasoning tasks through specialized reinforcement learning techniques that differ from standard RLHF approaches. A key distinction in training methodologies for LRMs is their reliance on reference-based reward systems, where rewards are assigned based on alignment between model-generated responses and authoritative reference answers. This approach

---
[*]Contribution during internship at Meituan Group.
[†]Corresponding authors.

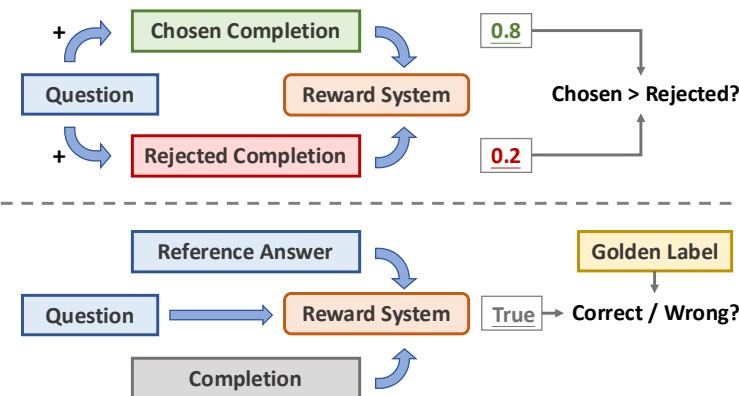

Figure 1: The core distinction between VerifyBench and existing reward benchmarks (Lambert et al., 2025; Liu et al., 2024) is illustrated as follows. **Upper panel:** Existing reward benchmarks assess the accuracy of a reward system by comparing the ranking of two completions for the same question. **Lower panel:** In contrast, our proposed VerifyBench evaluates the accuracy of a reward system by determining the correctness of a single completion using a reference answer.

has been implemented variously across leading models, with Deepseek-R1 employs a rule-based reward to prevent reward hacking, whereas models like Seed1.5-Thinking (Seed et al., 2025) adopt model-based reward systems to generate more precise and robust signals.

Despite the widespread adoption of reference-based reward systems in training state-of-the-art reasoning models, a significant gap exists in our ability to evaluate these systems systematically. Current benchmarks focus almost exclusively on preference-based evaluation, assessing rewards on their ability to rank competing responses correctly. This approach fails to capture the requirements of reference-based verification, where responses must be judged against objective ground truths rather than relative preferences (Kim et al., 2024). The absence of dedicated benchmarks for reference-based reward systems has limited researchers' ability to assess, compare, and improve their verification methodologies effectively, potentially impeding progress in reasoning model development.

To address this critical gap, we introduce VerifyBench, a benchmark specifically designed to evaluate the accuracy of reference-based reward systems. VerifyBench differs fundamentally from existing reward benchmarks by focusing on absolute correctness judgments rather than relative preference assessments. While traditional benchmarks ask reward models to determine which of two responses is better, VerifyBench challenges systems to verify whether a single response correctly aligns with a reference answer, more accurately reflecting the actual use case in reasoning model training.

In this paper, we present VerifyBench, a benchmark specifically designed to evaluate the accuracy of reference-based reward systems. To create VerifyBench, we curated a diverse collection of instructions paired with reference answers sourced from existing open datasets. Responses to these instructions were generated by multiple open-source and proprietary LLMs. The correctness of each response was assessed using both automated model judgments and human evaluations. Each instance in VerifyBench was verified by at least two human annotators to ensure label consistency and reliability, thereby producing a high-quality benchmark for the evaluation of reward systems.

Recognizing the need to differentiate between various verification techniques and to push the boundaries of current capabilities, we further developed VerifyBench-Hard, a more challenging variant of our benchmark. This dataset focuses on contentious cases where leading models produce highly conflicting judgments, providing a more stringent test for reward system accuracy. VerifyBench-Hard samples were carefully selected based on disagreement patterns among high-performing models, then subjected to thorough human annotation to ensure label quality.

Our contributions are summarized as follows:

- To better reflect realistic reinforcement learning (RL) scenarios for reasoning models, we construct VerifyBench, a benchmark derived from existing models and datasets, to provide an objective evaluation of the accuracy of reference-based reward systems.

- We further develop VerifyBench-Hard, a more challenging benchmark curated from cases exhibiting high disagreement among multiple models. This dataset contains a larger proportion of difficult-to-verify samples, highlighting substantial potential for improvement in current models.

- We conduct a comprehensive empirical analysis of model performance on both VerifyBench and VerifyBench-Hard, offering actionable insights to advance the accuracy of reference-based reward systems and enhance RL training in reasoning tasks.

## 2 PRELIMINARIES

**Reference-free Reward Models** In reinforcement learning (RL) for large language models (LLMs), the reward model plays a crucial role by approximating real-world reward signals associated with model-generated outputs. A typical reward model takes as input a user's query $q$ along with the corresponding LLM-generated response $r$, and produces a reward signal, formally defined as:

$$reward = R_\varphi(q, r) \tag{1}$$

where $q$ represents the user's query, $r$ denotes the response generated by the LLM, and $\varphi$ encapsulates either the learned parameters of the reward model or the heuristic criteria used to evaluate the quality of the response given $q$ and $r$.

**Evaluation of Reference-free Reward Models** Generally, reward models produce scalar outputs whose scales can vary significantly across different implementations, complicating direct numerical comparisons. Consequently, current benchmarks evaluate reward models using a pairwise comparative approach. Formally, given a dataset $D$ comprising tuples $(q, r_w, r_l)$, where $q$ represents a user's query, and $r_w$ and $r_l$ denote two candidate responses with $r_w$ considered superior to $r_l$, the accuracy of a reward model is quantified as the proportion of instances in which the model correctly assigns a higher score to $r_w$ than to $r_l$. Mathematically, this accuracy metric is defined as:

$$\text{Accuracy} = \frac{1}{|D|} \sum_{(q, r_w, r_l) \in D} \mathbb{I}\left[R_\varphi(q, r_w) > R_\varphi(q, r_l)\right] \tag{2}$$

where $\mathbb{I}(\cdot)$ is the indicator function, and $R_\varphi$ denotes the reward model parameterized by $\varphi$.

**Reference-based Reward Models** With the emergence of advanced reasoning models such as DeepSeek-R1, reference-based reward systems have been integrated into reinforcement learning (RL) frameworks for large reasoning models (LRMs). These models require training on extensive datasets, which typically include authoritative reference answers. Consequently, the reward assignment task shifts towards evaluating the alignment between the model-generated outputs and their corresponding reference answers. Formally, this reward calculation can be expressed as:

$$reward = R_\varphi(q, gt, r) \tag{3}$$

where $q$ denotes the user-issued query, $gt$ denotes the ground-truth reference answer, $r$ represents the model-generated response, and $\varphi$ encapsulates either the learned parameters of the reward model or the established evaluation criteria used to assess the alignment among $q$, $gt$, and $r$.

**Evaluation of Reference-based Reward Models** In this paper, we propose a reference-based reward benchmark designed to systematically evaluate reward models within reinforcement learning (RL) frameworks for large reasoning models (LRMs). Unlike traditional reward evaluation benchmarks, which rely on pairwise comparisons, our approach leverages explicit reference answers to directly assess the correctness of individual model-generated responses. Concretely, given a dataset $D$ consisting of instances $(q, gt, r, y)$, where $q$ denotes the user-issued query, $gt$ represents the ground-truth reference answer, $r$ is the model-generated response, and $y$ is the binary correctness label assigned to the response, we evaluate the reward model by measuring its accuracy in correctly predicting these labels. Formally, the accuracy metric is defined as:

$$\text{Accuracy} = \frac{1}{|D|} \sum_{(q, gt, r, y) \in D} \mathbb{I}\left[\mathbb{E}(R_\varphi(q, gt, r)) = y\right] \tag{4}$$

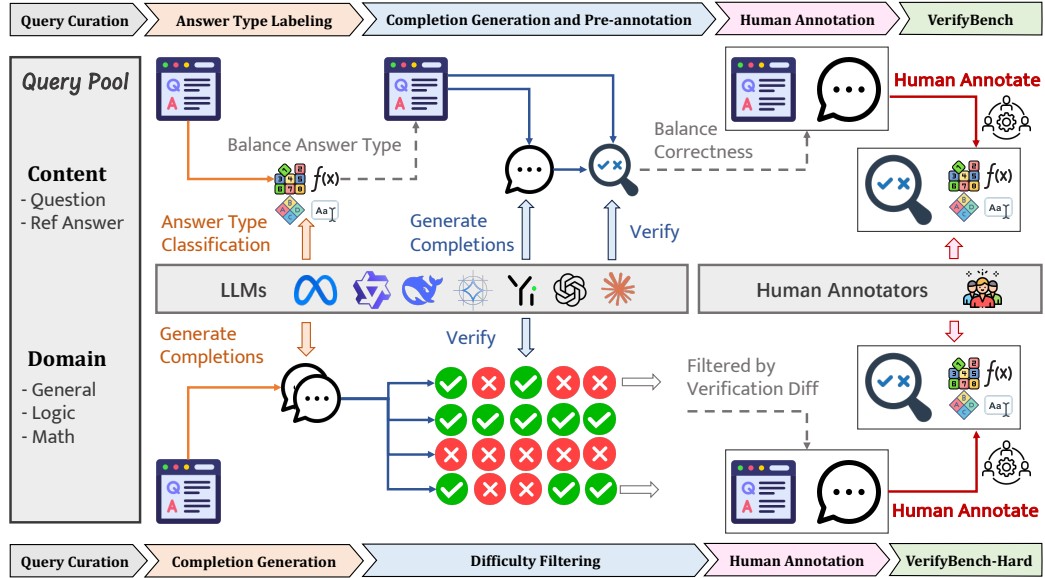

Figure 2: Overview of the benchmark construction process. The upper section outlines the pipeline used to construct VerifyBench, whereas the lower section details the pipeline for VerifyBench-Hard. The components highlighted by black boxes denote the final entries included in the benchmark.

where $R_\varphi(q, gt, r)$ denotes the reward model parameterized by $\varphi$ or defined by heuristic verification rules, producing predictions indicative of the correctness of response $r$ relative to the provided reference answer $gt$. The function $\mathbb{E}(\cdot)$ represents an operation (e.g., thresholding or discretization) mapping continuous reward scores into discrete correctness predictions suitable for direct comparison with the ground-truth labels $y$.

## 3 BENCHMARK CONSTRUCTION

In this paper, we introduce two benchmarks, VerifyBench and VerifyBench-Hard, to evaluate reference-based reward systems. The VerifyBench benchmark (Section 3.1) is designed to reflect naturally distributed data, whereas VerifyBench-Hard (Section 3.2) comprises samples exhibiting high levels of disagreement among models, thereby assessing a model's ability to provide reliable judgments in ambiguous or challenging scenarios.

### 3.1 CONSTRUCTION OF VERIFYBENCH

**Query Curation** To emulate realistic reinforcement learning (RL) scenarios involving reference-based reward systems, we curate a comprehensive collection of open-source reasoning problems paired with corresponding reference answers. These problems encompass three primary categories, general reasoning, logical reasoning and mathematical reasoning, and are aggregated from 41 distinct sources. A complete list of these data sources is provided in Appendix D.

**Answer Type Labeling** To comprehensively evaluate model performance across diverse answer formats, we define four canonical answer types: numerical values, algebraic expressions, multiple-choice selections, and free-form strings. Utilizing a general-purpose LLM Llama-3.3-70B-Instruct (Grattafiori et al., 2024), we performed automatic answer-type classification with a prompt (Appendix G.1). Questions that fall outside these categories, such as proof-based or open-ended prompts, were excluded from further analysis. Following classification, we randomly sampled 2,000 instances per answer type, resulting in a final candidate pool of 8,000 questions.

**Completion Generation and Pre-annotation** We subsequently employed 22 widely used open-source and proprietary models (see Appendix E) to generate single-shot completions for the curated

set of 8,000 questions, resulting in a total of 176,000 completions. To assign initial correctness labels, we utilized Llama-3.3-70B-Instruct (Grattafiori et al., 2024) within a prompt-based judgment framework. For each question, we randomly selected four completions, two labeled as correct and two labeled as incorrect by the model, and retained them for subsequent human annotation.

**Human Annotation**   We conducted human annotation for the aforementioned questions and their associated completions. The annotation procedure comprised two primary tasks: (1) identifying the most appropriate answer type for each question based on its formulation and corresponding ground-truth answer, and (2) evaluating the correctness of each of the four completions. Each question was independently annotated by at least two annotators. If their annotations were consistent, the labeling was finalized; otherwise, a meta-annotator resolved disagreements to ensure consistency and finalize the labels. A detailed description of the human annotation process is provided in Appendix H .

**Benchmark Construction**   Following human annotation, we identified notable biases in the models' predictions regarding both answer types and completion correctness, leading to imbalanced data distributions. To mitigate this issue, we performed controlled downsampling to ensure uniform category-level representation and balanced correctness labels. Specifically, we retained 250 questions per answer type, resulting in a total of 1,000 questions. Each question is paired with exactly two completions, one correct and one incorrect, aimed to minimize bias arising from class imbalance. In this way, we maintain an equal number of samples across these 4 answer types in VerifyBench, enabling us to assess a model's verification accuracy under different scenarios. The resulting dataset, VerifyBench, thus comprises 2,000 well-balanced question-answer-completion-correctness tuples. Detailed statistics for VerifyBench are provided in Table 1 and Appendix F, and illustrative examples are presented in Appendix K and  L.

## 3.2   CONSTRUCTION OF VERIFYBENCH-HARD

To construct VerifyBench-Hard, we employed a specialized data generation pipeline consisting of the following key steps:

**Completion Generation**   To construct the dataset, we first generated single-turn completions for the queries described in Section 3.1 using a collection of 18 open-source models (Appendix E). Due to the substantial volume of generations and the associated computational costs, closed-source models were excluded from this stage. In total, we produced approximately 1.45 million completions.

**Difficulty Filtering**   Next, we employed five top-performing LLMs on VerifyBench (Llama-3.3-70B-Instruct (Grattafiori et al., 2024), Llama-4-Scout-17B-16E-Instruct (Meta AI, 2025), Qwen2.5-72B-Instruct (Qwen et al., 2025), Qwen3-30B-A3B, and Qwen3-32B (Yang et al., 2025)), which span a diverse range of model sizes and architectures, to

Table 1: Benchmark statistics of VerifyBench (VB) and VerifyBench-Hard (VB-H).

| Statistics for VerifyBench and VerifyBench-Hard | | |
|---|---|---|
| **Statistics** | **VB** | **VB-H** |
| # of unique questions | 1000 | 945 |
| # of unique completions | 2000 | 1000 |
| # of correct completions | 1000 | 291 |
| # of wrong completions | 1000 | 709 |
| *Answer Type Distribution* | | |
| # of Numeric Values | 500 | 252 |
| # of Expressions | 500 | 88 |
| # of Multi-choice | 500 | 430 |
| # of String | 500 | 230 |
| *Domain Distribution* | | |
| # of General Reasoing | 404 | 303 |
| # of Logic Reasoning | 498 | 315 |
| # of Math Reasoning | 1098 | 382 |

evaluate the correctness of the generated completions. Based on their judgments, we identified question-answer-completion tuples exhibiting model disagreement, specifically those for which two models' assessments diverged from the other three. To ensure balanced and comprehensive representation, we applied stratified sampling across data domains and sources, ultimately selecting 2,000 examples for human annotation.

**Human Annotation**   We subsequently subjected the selected samples to human annotation, focusing on two key aspects: identifying the answer type and determining the correctness of each completion. Each instance was annotated independently by at least two annotators. In cases where

Table 2: Overall performance(%) of VerifyBench and VerifyBench-Hard. **Num** stands for Numeric Values, **Exp** stands for Expressions, **MC** stands for Multi-choice and **Str** stands for String.

| Model/Method | VerifyBench | | | | | VerifyBench-Hard | | | | |
|---|---|---|---|---|---|---|---|---|---|---|
| | Num | Exp | MC | Str | AVG | Num | Exp | MC | Str | AVG |
| *Rule-based functions* | | | | | | | | | | |
| math-verify | 85.60 | 75.60 | 55.00 | 51.60 | 66.95 | 84.52 | 82.95 | 68.37 | 78.26 | 76.00 |
| *LLM-as-a-judge* | | | | | | | | | | |
| gpt-4o-2024-11-20 | 94.80 | 90.20 | 96.80 | 90.80 | 93.15 | 71.43 | 65.91 | 75.35 | 71.30 | 72.60 |
| gpt-4o-mini | 95.80 | 89.80 | 95.80 | 90.00 | 92.85 | 69.05 | 72.73 | 74.19 | 72.17 | 72.30 |
| DeepSeek-V3-0324 | 96.80 | 93.00 | 97.60 | 91.60 | 94.75 | 81.35 | 77.27 | 81.63 | 75.22 | 79.70 |
| DeepSeek-R1-0528 | **98.00** | 92.60 | 98.00 | 92.00 | 95.15 | 82.14 | **81.82** | 90.93 | 85.22 | 86.60 |
| gpt-oss-20b | 97.40 | 94.20 | 98.40 | 89.20 | 94.80 | 82.54 | 79.55 | 90.23 | 76.96 | 84.30 |
| gpt-oss-120b | **98.00** | **94.80** | **99.20** | 91.40 | **95.85** | **84.13** | 80.68 | **92.56** | **86.09** | **87.90** |
| Llama-4-17B-16E-Instruct | 94.20 | 86.80 | 89.80 | 89.25 | 90.01 | 48.02 | 39.77 | 46.98 | 55.22 | 48.50 |
| Llama-3.3-70B-Instruct | 88.80 | 77.80 | 88.40 | 78.00 | 83.25 | 54.37 | 45.45 | 60.70 | 47.39 | 54.70 |
| Llama-3.1-8B-Instruct | 72.20 | 70.60 | 77.00 | 72.40 | 73.05 | 51.19 | 35.23 | 45.12 | 33.91 | 43.20 |
| Llama-3.2-3B-Instruct | 65.80 | 63.60 | 56.80 | 57.60 | 60.95 | 33.33 | 28.41 | 38.84 | 27.39 | 33.90 |
| Llama-3.2-1B-Instruct | 44.40 | 41.00 | 37.60 | 53.60 | 44.15 | 22.22 | 13.64 | 29.07 | 27.39 | 25.60 |
| Qwen3-235B-A22B | 96.40 | 92.40 | 97.00 | 89.40 | 93.80 | 70.24 | 72.73 | 70.93 | 69.57 | 70.60 |
| Qwen3-30B-A3B | 96.60 | 91.80 | 97.40 | 90.20 | 94.00 | 64.68 | 70.45 | 69.53 | 56.52 | 65.40 |
| Qwen2.5-72B-Instruct | 95.40 | 89.80 | 95.60 | 88.60 | 92.35 | 70.63 | 60.23 | 61.40 | 56.09 | 62.40 |
| Qwen3-32B | 97.60 | 94.00 | 99.00 | **92.60** | 95.80 | 69.05 | **81.82** | 68.14 | 77.83 | 71.80 |
| Qwen3-8B | 96.40 | 93.00 | 96.20 | 90.40 | 94.00 | 68.65 | 78.41 | 73.02 | 66.52 | 70.90 |
| Qwen3-4B | 95.20 | 91.60 | 93.60 | 87.60 | 92.00 | 71.03 | 62.50 | 75.58 | 71.74 | 72.40 |
| Qwen3-1.7B | 83.20 | 81.00 | 80.60 | 79.60 | 81.10 | 48.81 | 38.64 | 60.93 | 41.74 | 51.50 |
| phi-4 | 92.60 | 86.40 | 93.00 | 85.40 | 89.35 | 59.52 | 57.95 | 54.19 | 57.39 | 56.60 |
| Yi-1.5-9B-Chat-16K | 90.40 | 87.40 | 88.00 | 85.00 | 87.70 | 65.48 | 63.64 | 62.09 | 54.78 | 61.40 |
| gemma-3-1b-it | 55.40 | 56.20 | 43.00 | 56.00 | 52.65 | 32.14 | 19.32 | 33.72 | 40.87 | 33.70 |
| *Model-based Verifier* | | | | | | | | | | |
| xVerify-0.5B-I | 79.60 | 74.40 | 88.80 | 87.20 | 82.50 | 60.32 | 62.50 | 91.86 | 77.83 | 78.10 |
| xVerify-3B-Ia | 89.80 | 86.40 | 95.00 | 90.20 | 90.35 | 73.81 | 78.41 | 93.49 | 73.91 | 82.70 |
| xVerify-8B-I | 93.80 | 89.60 | 95.20 | 89.20 | 91.95 | 69.05 | 76.14 | 93.49 | 81.74 | 83.10 |
| Tencent-Qwen2.5-7B-RLVR | **96.80** | **95.40** | 93.00 | **90.80** | **94.00** | 75.43 | 75.00 | 61.36 | 63.64 | 72.20 |
| CompassVerifier-3B | 92.00 | 84.80 | 94.60 | 88.60 | 90.00 | 77.38 | 84.09 | 93.95 | 75.22 | 84.60 |
| CompassVerifier-7B | 92.80 | 88.60 | **97.20** | 88.40 | 91.75 | 78.97 | **85.23** | 95.35 | 76.09 | 85.90 |
| CompassVerifier-32B | 94.20 | 89.80 | 96.80 | 89.40 | 92.55 | **80.56** | 82.95 | **95.58** | **83.48** | **87.90** |

both annotators agreed, the annotation was finalized; when disagreement occurred, a meta-annotator was consulted to resolve the conflict. A detailed description of the human annotation process is provided in Appendix H.

**Benchmark Construction**    Following human annotation, we excluded samples identified as unsuitable for inclusion in our benchmark. This filtering resulted in a final set of 1,000 question-answer-completion-correctness tuples. In contrast to VerifyBench, which enforces a balanced structure with one correct and one incorrect completion per question, VerifyBench-Hard is derived through natural sampling. We observed that larger models are more likely to erroneously accept incorrect answers as correct, resulting in a natural skew towards incorrect completions within the dataset. Detailed statistics for VerifyBench-Hard are provided in Table 1.

## 4    EVALUATION RESULTS

This section presents the evaluation results and analyses of our proposed benchmark. Section 4.1 reports the primary evaluation outcomes. In Section 4.2, we investigate the impact of reference answers on the verification process. Section 4.3 provides a comparative analysis between our benchmark and existing reward benchmarks, as well as the performance of several general-purpose reward models on VerifyBench and VerifyBench-Hard.

## 4.1 Overall Performance

We evaluate the performance of various verification approaches on both VerifyBench and VerifyBench-Hard. For rule-based baselines, we adopt the widely used *math-verify* (Kydlíček, 2025) method[1]. In the LLM-as-a-judge setting, we prompt general LLMs to perform verification; detailed prompt templates are provided in Appendix G.2. Additionally, we conduct evaluations of existing verification-specialized model-based verifiers, which have been carefully trained on verification tasks, employing prompts consistent with their provided templates to ensure fairness (Chen et al., 2025; Su et al., 2025; Liu et al., 2025a). Our evaluation yields several key findings:

**Existing models perform well on VerifyBench.** The primary objective in constructing Verify-Bench is to establish a benchmark for the objective evaluation of reference-based reward systems. To this end, we designed the dataset with a balanced distribution across diverse domains and answer types, pairing each question with both a correct and an incorrect completion. This structure facilitates a rigorous and fair assessment of reward model performance. Notably, state-of-the-art LLMs already demonstrate strong performance on this benchmark: GPT-4o-mini achieves an average accuracy of 92.85%, while Qwen3-32B reaches 95.8%, highlighting the high reliability of LLMs as verifiers in this context.

**VerifyBench-Hard is challenging.** To more effectively differentiate the performance of various models, we constructed VerifyBench-Hard by selecting cases in which multiple LLMs exhibited substantial disagreement in their verification outputs. Evaluation results demonstrate that model performance on VerifyBench-Hard is significantly lower than on VerifyBench. The highest accuracy achieved is 72.4%, representing a 20% decrease compared to performance on VerifyBench. This performance gap underscores substantial opportunities for improvement in the precise verification capabilities of current LLMs.

**Small-scale models still have room for development.** In practical reinforcement learning scenarios, the inference efficiency of the reward system significantly impacts the overall training speed. Since such verification tasks typically involve generative inference, their computational cost is comparable to that of the rollout process itself. Thus, efficiently leveraging smaller models to perform verification is a practical concern worth exploring. According to our results, models with smaller parameters ($<$3B parameters) exhibit notably poorer performance on VerifyBench, achieving 81.10% accuracy with Qwen3-1.7B and only 60.95% accuracy with Llama-3.2-3B-Instruct, while larger-scale models can achieve over 90% accuracy. Therefore, enhancing the capability of smaller models on these verification tasks represents a valuable direction for future research.

## 4.2 Reference-answers Play an Important Role in Verification

The benchmark proposed in this work fundamentally differs from existing reward benchmarks by explicitly incorporating reference answers, thereby aligning more closely with the training setups of contemporary reasoning LLMs. To isolate the impact of reference answers on verification performance, we conduct an ablation study in which models are evaluated without reference answers provided in the prompt; the prompt format used is detailed in Appendix G.3.

Experimental results, summarized in Table 3, reveal a performance degradation of approximately 5-18% when reference answers are excluded. These findings underscore the crucial role of reference answers in reasoning-oriented

Table 3: Evaluation results(%) about how including the reference answer in the prompt influences the performance of LLM-as-a-judge.

| Model | VerifyBench | |
|---|---|---|
| | w/ Ref | w/o Ref |
| Llama-4-Scout-17B-16E-Instruct | 90.01 | 73.95-16.06 |
| Llama-3.3-70B-Instruct | 83.25 | 75.00-8.25 |
| Llama-3.1-8B-Instruct | 73.05 | 64.10-8.95 |
| Llama-3.2-3B-Instruct | 60.95 | 55.35-5.60 |
| Llama-3.2-1B-Instruct | 44.15 | 44.50+0.35 |
| Qwen3-235B-A22B | 93.80 | 80.15-13.65 |
| Qwen3-30B-A3B | 94.00 | 78.25-15.75 |
| Qwen2.5-72B-Instruct | 92.35 | 77.30-15.05 |
| Qwen3-32B | 95.80 | 78.90-16.90 |
| Qwen3-8B | 94.00 | 75.75-18.25 |
| Qwen3-4B | 92.00 | 74.40-17.60 |
| Qwen3-1.7B | 81.10 | 62.10-19.00 |

---

[1]For math-verify, we report the accuracy of version v0.8.0.

Table 4: The performance(%) of existing reward models on VerifyBench without access to reference answers, as well as a comparison with existing reward benchmarks.

| Model | RM-Bench | Reward Bench | VerifyBench | | | | |
|---|---|---|---|---|---|---|---|
| | | | Num | Exp | MC | Str | AVG |
| *General Reward Models* | | | | | | | |
| Skywork-Reward-Llama-3.1-8B | 72.29 | **93.33** | 60.80 | 64.80 | 59.60 | 68.80 | 63.48 |
| internlm2-20b-reward | 72.06 | 92.16 | 65.60 | 64.80 | 61.20 | 70.00 | 65.40 |
| GRM-llama3-8B-sftreg | 71.33 | 88.50 | 64.80 | 58.40 | 58.80 | 67.60 | 62.40 |
| internlm2-7b-reward | 72.42 | 90.02 | 73.20 | 68.00 | 66.80 | 70.40 | 69.60 |
| *Domain-specific Reward Models* | | | | | | | |
| Qwen2.5-Math-RM-72B | **76.28** | 82.11 | **83.60** | **79.20** | **73.60** | **75.60** | **78.00** |
| Qwen2-Math-RM-72B | 62.61 | 75.54 | 79.20 | 78.40 | 73.20 | 72.80 | 75.90 |

RL, suggesting they provide a more reliable and informative supervision signal during reward modeling.

## 4.3 PERFORMANCE OF REFERENCE-FREE REWARD MODELS

To enable a more comprehensive evaluation of existing reward models, we additionally assessed several reference-free reward models and benchmarked their performance on conventional pairwise reward evaluation datasets for comparison. Notably, each question in our proposed VerifyBench consists of one correct and one incorrect completion, enabling straightforward reformulation into standard pairwise evaluation instances. The experimental results are summarized in Table 4.

Our experimental results show that VerifyBench introduces a level of challenge comparable to existing reward benchmarks, with the absence of reference answers. Reference-free reward models achieve sub-80% accuracy on VerifyBench, highlighting its difficulty. Furthermore, domain-specific reward models exhibit inferior performance on general reward benchmarks (Liu et al., 2024; Lambert et al., 2025) compared to VerifyBench, validating the benchmark's design objectives.

## 5 ANALYSIS

### 5.1 ERROR ANALYSIS

To gain deeper insights from VerifyBench, we introduce a more fine-grained taxonomy for each answer type and analyze model performance across these subcategories. This detailed analysis helps identify specific reasoning tasks or answer formats where models are particularly error-prone. We subdivide the Numeric Values category into 8 subtypes, Expressions into 5 subtypes, Multichoice into 3 subtypes, and String into 2 subtypes. Table 5 presents the comparative performance of different models across these detailed categories. The complete taxonomy of VerifyBench and VerifyBench-Hard is provided in Appendix I.

We further analyze subcategories within each major category that exhibit below-average accuracy. The following error-prone subtypes are identified as the most frequent sources of incorrect judgments:

- **Numeric Values:** Complex numbers and answers containing multiple numerical values;
- **Expressions:** Formulas and equations;
- **Multi-choice:** Multiple-choice problems;
- **String:** Strings requiring semantic consistency verification.

We analyzed the samples most prone to errors and identified a common underlying issue: models frequently fail to fully comprehend the question or clearly recognize the intended objective. For instance, in cases involving multi-value answers, the ordering of values is typically irrelevant. However, if the sequence of values in the model's output differs from the golden answer, models

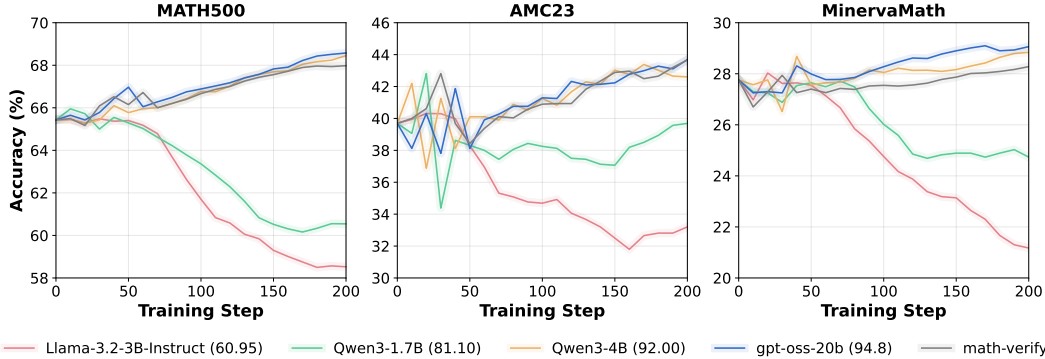

Figure 3: The performance(%) of RL across different LLM judges which have various performance on VerifyBench.

often incorrectly classify the response as erroneous. Similarly, errors within the Expressions category, particularly involving algebraic formulas and equations, predominantly result from inadequate mathematical comprehension. Specifically, when a model outputs an unsimplified expression, superficial textual discrepancies compared to the ground-truth answer can be significant. Rather than evaluating whether the expression is mathematically equivalent upon simplification, models prematurely deem the output incorrect, thereby leading to verification failures. We provide, in Appendix M, examples of misjudgments made by gpt-oss-120b, the best-performing open-source model on VerifyBench within each category. These examples span different answer types and are intended to help better understand why LLM verifiers make errors.

## 5.2 CORRELATION ANALYSIS

We constructed VerifyBench and VerifyBench-Hard with the goal of improving the effectiveness of RL for reasoning models by enhancing the accuracy of reference-based reward systems. To evaluate the practical utility of our benchmark, we performed a correlation analysis between VerifyBench and real-world RL performance.

We conducted practical RL training on Qwen2.5-3B-Instruct (Qwen et al., 2025), using LLMs with varying performance on VerifyBench as LLM judgers to serve as the reward system. We then compared their improvements on downstream tasks. Specifically, we adopted math-verify (Kydlíček, 2025), Llama-3.2-3B-Instruct (Grattafiori et al., 2024), Qwen3-1.7, Qwen3-4B (Yang et al., 2025), and gpt-oss-20b (OpenAI et al., 2025) as verifiers, employed GRPO (Shao et al., 2024) as the RL algorithm, and trained on DeepMath-103K. The accuracy changes on MATH500 (Hendrycks et al., 2021), AMC23 (mathai, 2023), and MinervaMath (Lewkowycz et al., 2022) are shown in Figure 3. Meanwhile, we provide a additional correlation analysis based on Rejection Sampling Fine-tuning and all the experimental details in Appendix J.

Table 5: Model performance(%) across the fine-grained taxonomy on VerifyBench. **Q32B** stands for Qwen3-32B, **g4o** stands for gpt-4o-2024-11-20, **L70B** stands for Llama-3.3-70B-Instruct and **L3B** stands for Llama-3.2-3B-Instruct.

| Answer Type | Q32B | g4o | L70B | L3B |
|---|---|---|---|---|
| Numeric Values | 97.60 | 94.80 | 88.80 | 65.80 |
| Integer | 96.88-0.72 | 96.88 | 93.75 | 65.62-0.18 |
| Constant | 96.88-0.72 | 95.31 | 92.19 | 70.31 |
| Float | 98.39 | 96.77 | 90.32 | 61.29-4.51 |
| Radical | 98.39 | 95.16 | 87.10-1.70 | 75.81 |
| Complex | 96.77-0.83 | 96.77 | 85.48-3.32 | 59.68-6.12 |
| Angle | 96.77-0.83 | 96.77 | 93.55 | 66.13 |
| Non-decimal | 100 | 93.55-1.25 | 88.71-0.09 | 64.52-1.28 |
| Multiple values | 96.77-0.83 | 87.10-7.70 | 79.03-9.77 | 62.90-2.90 |
| Expressions | 94.00 | 90.20 | 77.80 | 63.60 |
| Formula | 91.54-2.46 | 84.62-5.58 | 67.69-10.11 | 56.92-6.68 |
| Equation | 87.50-6.5 | 78.12-12.08 | 70.31-7.49 | 60.94-2.66 |
| Interval | 96.09 | 94.53 | 82.81 | 60.94-2.66 |
| Set | 98.00 | 98.00 | 78.00 | 60.00-3.60 |
| Matrix | 96.09 | 94.53 | 86.72 | 75.78 |
| Multi-choice | 99.00 | 96.80 | 88.40 | 56.80 |
| Single-choice | 99.39 | 98.17 | 92.07 | 59.15 |
| Multiple-choice | 98.21-0.79 | 94.05-2.75 | 77.98-10.42 | 49.40-7.40 |
| State selection | 99.40 | 98.21 | 95.24 | 61.90 |
| String | 92.60 | 90.80 | 78.00 | 57.6 |
| Specific | 93.60 | 93.31 | 81.69 | 59.01 |
| Semantic | 90.38-2.22 | 85.26-5.54 | 69.87-8.13 | 54.49-3.11 |

Sampling Fine-tuning and all the experimental details in Appendix J.

From our experimental results, we observe that models performing well on VerifyBench, when used as verifiers in RL training, lead to greater improvements on related tasks. For example, employing

gpt-oss-20b and Qwen3-4B yields significant gains, whereas using models with weaker VerifyBench performance as verifiers may even degrade the model's capability. This is because such models introduce substantial bias in reward assignment, resulting in inaccurate supervision. Our experiments demonstrate that VerifyBench provides effective guidance in RL training scenarios, highlighting the practical utility of our benchmarks.

# 6 CONCLUSION

In this paper, we present two dedicated benchmarks, VerifyBench and VerifyBench-Hard, to evaluate reference-based reward systems in the context of reasoning-focused reinforcement learning. These benchmarks were built with high-quality, carefully curated data and extensive human annotation. Our results reveal that current verifiers, especially those with smaller model sizes, still face considerable challenges in accurately assessing reasoning completions. Through detailed analysis, we provide insights into the strengths and weaknesses of existing systems and highlight opportunities for improvement. The proposed benchmarks fill a critical gap in the evaluation landscape, offering a principled foundation for understanding verifier accuracy and guiding the development of more effective reasoning models trained via reinforcement learning.

## ACKNOWLEDGEMENT

This work was supported by National Natural Science Foundation of China (No. 62506332), "Pioneer" and "Leading Goose" R&D Program of Zhejiang (NO. 2026C02A1223) and CCF-Tencent Rhino-Bird Open Research Fund.

## ETHICS STATEMENT

All human annotators involved in constructing the benchmarks were assigned reasonable workloads and fairly compensated for their contributions.

Our annotation process involves minimal subjective preference. Human annotators performed the verification tasks following our detailed instructions. The content of the annotations does not involve ethical issues and poses no ethical risks.

## REPRODUCIBILITY STATEMENT

We provide the VerifyBench dataset and evaluation code in the supplementary materials to facilitate the research community's use of our benchmark. In Appendix J, we detail the hyperparameter settings used in our evaluations, enabling researchers to reproduce our results. For the RL experiments in Section 5.2, we conducted experiments using the 'remote_rm' recipe provided by veRL (Sheng et al., 2025). The detailed experimental configurations are also included in Appendix J to support reproducibility. Nevertheless, due to the inherent stochasticity in LLM training and inference, results may still exhibit variability.

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

CONTENTS

## A    LLM Usage Declaration

In writing this paper, we only used LLMs for polishing. The generation of ideas in this work **did not** involve any assistance from LLMs. The experimental design and manuscript writing were **not directly produced by LLMs** either. The models were used solely as a polishing tool: specifically, we first drafted the manuscript, then refined it with the help of an LLM, and finally the authors conducted another round of verification after polishing.

## B    Limitations

**Limited Data Domain**    In this paper, we utilize datasets exclusively from general reasoning, logical reasoning, and mathematical reasoning, which do not cover the full spectrum of reasoning types, such as commonsense reasoning. Consequently, our test sets may not adequately evaluate the quality of reward systems in out-of-domain scenarios.

**Bias from Human Annotation**    The construction of VerifyBench and VerifyBench-Hard involved extensive human annotation. Although all annotators were trained and a double-checking strategy was employed, it remains challenging to entirely eliminate annotation bias inherent in manual labeling processes.

**Reward Hacking Could Not Be Identified**    While our experiments demonstrate that rule-based reward systems perform worse than model-based approaches on both VerifyBench and VerifyBench-Hard, a critical issue remains unaddressed: reward hacking. Future research should focus on detecting and evaluating reward hacking phenomena.

**Proof Problems Excluded**    During annotation, our guidelines explicitly excluded proof-based questions. We believe such problems require more specialized verification methods, such as formal languages like Lean4. Consequently, proof questions are not included in this study, and their verification remains an open research challenge.

**Binary Scoring System**    The benchmark constructed in this paper employs a binary scoring system, where each completion is labeled as either correct or incorrect. However, real-world scenarios often involve more nuanced cases, such as partially correct reasoning processes or correct solutions to subproblems. Introducing a more fine-grained evaluation scheme could better capture these complexities.

## C    Related Works

### C.1    Reward Systems for Reinforcement Learning

Early reward models (RMs) (Christiano et al., 2017; Stiennon et al., 2020; Ouyang et al., 2022), trained to predict human preference rankings, typically treat the entire response as the evaluation unit. However, such outcome-level RMs lack insight into intermediate reasoning steps, making step-level error correction infeasible (Xu et al., 2025b). To address this limitation, process-level RMs (Lightman et al., 2023; Setlur et al., 2024) have been introduced to assign scores at each reasoning step, thereby providing stepwise feedback. Despite their effectiveness, process-level RMs require extensive manual step-level annotations, resulting in exponential increases in data collection costs and training complexity (Khalifa et al., 2025).

Building on these advances, DeepSeek-R1 (DeepSeek-AI et al., 2025a) employs rule-based reward functions that leverage predefined, maintainable rules for pattern matching and logical validation, offering simplicity and efficiency. However, as task diversity expands, the manual creation of such rules faces significant challenges related to scalability and coverage, ultimately limiting its applicability in open-ended generation scenarios.

More recently, DeepSeek-GRM (Liu et al., 2025b) and ThinkPRM (Khalifa et al., 2025) have explored integrating reasoning capabilities into RMs by developing generative reward models (GRMs). GRMs reformulate the scoring task as a token-generation problem: before outputting a numerical

score, the model first generates a chain-of-thought (CoT) (Wei et al., 2022) that explicates its evaluation criteria and rationale. This approach not only bridges the interpretability gap between black-box discriminative models and brittle rule-based systems but also substantially enhances test-time scaling capabilities.

With the advancement of Reinforcement Learning with Verifiable Reward (RLVR), using reference answers to verify correctness on reasoning tasks has become the mainstream approach for RL training of reasoning models. Most existing work employs rule-based verifiers to assess the consistency between rollouts and reference answers (DeepSeek-AI et al., 2025a; He et al., 2025), thereby providing rewards in RL. These rule-based functions typically rely on extensive human-crafted heuristic rules and often lack verification robustness (Hong et al., 2025). Consequently, recent work has incorporated reference answers into reward models, leveraging the generalization capabilities of model-based verifiers to improve verification accuracy (Chen et al., 2025; Liu et al., 2025a; Su et al., 2025).

## C.2 EVALUATION OF REWARD SYSTEMS

There are two primary approaches to evaluating reward systems. The first approach employs standardized benchmarks that objectively assess reward system effectiveness by designing diverse tasks and datasets (Frick et al., 2024). The second approach examines the performance of reward systems when integrated directly into downstream optimization loops, such as Best-of-N selection (Nakano et al., 2022) or rejection sampling fine-tuning (Zelikman et al., 2024; Xiong et al., 2025), to measure their impact on generation quality and alignment.

Reward system benchmarks can be further categorized into outcome-level (Liu et al., 2024; Lambert et al., 2025) and process-level (Lightman et al., 2023; Zheng et al., 2025) suites. In constructing these benchmarks, researchers generate multiple responses to the same prompt by varying model architectures or hyperparameters. During the manual annotation phase, outcome-level benchmarks require annotators to compare or assign multi-point scores to complete responses, emphasizing overall preference. In contrast, process-level benchmarks provide fine-grained gold verdicts by requiring step-by-step correctness labels for each reasoning step.

Beyond benchmark-based evaluation, practical applications of reward systems serve as another common assessment method. In the Best-of-N (BoN) paradigm, WebGPT (Nakano et al., 2022) introduced using a reward model to score $N$ candidate answers and select the top-ranked response. Subsequent work has framed reward models as downstream rankers, for example, Self-Consistency in chain-of-thought models (Wang et al., 2022), where the reward model identifies the most coherent solution among candidates. Unlike Best-of-N, rejection sampling fine-tuning (RFT) (Zelikman et al., 2024; Xiong et al., 2025) samples multiple trajectories from the current policy, scores them using a reward model, and retains only the highest-scoring examples as silver supervision for further fine-tuning. This approach has proven particularly effective at bootstrapping reasoning capabilities without requiring full preference-learning pipelines.

During the course of this work, we also discovered several concurrent works that constructed evaluation benchmarks for reward systems based on reference answers. Specifically, Liu et al. (2025a) proposed VerifierBench, which employs a similar construction approach to our VerifyBench (easy version). However, compared to our work, they utilized fewer datasets to generate evaluation samples, using 14 data sources while we employed 41 data sources. Additionally, we adopted a more fine-grained classification system for annotation. Furthermore, we propose a disagreement-based sample selection strategy to construct VerifyBench-Hard, which includes samples that are difficult for model-based verifiers to evaluate, thereby enhancing the benchmark's difficulty. Our benchmarks also incorporate outputs from reasoning models, aligning with the mainstream forms of contemporary model outputs and improving the benchmarks' coverage.

## D DATA SOURCE

Table 6 provides a comprehensive overview of all datasets used in constructing VerifyBench, detailing their respective licenses and the number of samples drawn from each. All data usage strictly complies with the terms and conditions stipulated by the original sources.

Table 6: The datasets we used and the number of samples drawn from each, including the license information of these datasets.

| Domain | Source | License | # of Questions |
|---|---|---|---|
| general_reasoning | BBH (Suzgun et al., 2023) | MIT | 4520 |
| | BBEH (Kazemi et al., 2025) | Apache 2.0 | 6511 |
| | MMLU_pro (Wang et al., 2024) | Apache 2.0 | 2000 |
| | natural_reasoning (Yuan et al., 2025) | CC-BY-NC 4.0 | 10000 |
| logic_reasoning | AbductionRules (Young et al., 2022) | MIT | 1000 |
| | anlg (Bhagavatula et al., 2019) | / | 1000 |
| | anli (Nie et al., 2020) | CC-BY-NC 4.0 | 1000 |
| | ARLSAT (Zhong et al., 2021) | MIT | 230 |
| | bAbI15 (Weston et al., 2015) | / | 1000 |
| | bAbI16 (Weston et al., 2015) | / | 1000 |
| | BoardgameQA (Kazemi et al., 2023) | CC-BY-4.0 | 1000 |
| | clutrr (Sinha et al., 2019) | CC-BY-NC 4.0 | 1000 |
| | FOLIO (Han et al., 2024) | CC-BY-SA-4.0 | 134 |
| | hellaswag (Zellers et al., 2019) | MIT | 1000 |
| | logicbenchBQA (Parmar et al., 2024) | MIT | 1000 |
| | logicbenchMCQA (Parmar et al., 2024) | MIT | 1000 |
| | LogiQA (Liu et al., 2020) | / | 1000 |
| | MultiLogiEval (Patel et al., 2024) | MIT | 1000 |
| | NeuLRabductive (Xu et al., 2025a) | / | 1000 |
| | NeuLRdeductive (Xu et al., 2025a) | / | 1000 |
| | NeuLRinductive (Xu et al., 2025a) | / | 1000 |
| | ProntoQA (Saparov & He, 2022) | Apache 2.0 | 500 |
| | ProofWriter (Tafjord et al., 2021) | / | 1000 |
| | ReClor (Yu et al., 2019) | / | 500 |
| | tablebench (Wu et al., 2025) | Apache 2.0 | 886 |
| math_reasoning | AIME24 | MIT | 30 |
| | AIME25 | MIT | 30 |
| | asdiv-a (Miao et al., 2020) | CC-BY-NC 4.0 | 1218 |
| | Math Odyssey (Fang et al., 2025) | MIT | 389 |
| | GPQA_diamond (Rein et al., 2024) | MIT | 198 |
| | gsm8k (Cobbe et al., 2021) | MIT | 1319 |
| | math401 (Yuan et al., 2023) | / | 392 |
| | mathematics (Saxton et al., 2018) | Apache 2.0 | 3360 |
| | MATH(Hendrycks et al., 2021) | MIT | 5000 |
| | OlympiadBench-EN (He et al., 2024) | MIT | 675 |
| | SVAMP (Patel et al., 2021) | MIT | 1000 |
| | NuminaMath-CoT (Li et al., 2024) | Apache 2.0 | 20000 |
| | orca-math-word-problems (Mitra et al., 2024) | MIT | 10000 |
| | ArtOfProblemSolving | self-curated | 7997 |
| | stackmathqa (Zhang, Yifan, 2024) | CC-BY-4.0 | 10000 |
| | DeepMath-103K-RL (He et al., 2025) | MIT | 20000 |

The datasets we used mainly fall into three categories: *general_reasoning*, *logic_reasoning*, and *math_reasoning*. Among them, *general_reasoning* also includes subsets such as *commonsense_reasoning*. In our selection, *general_reasoning* consists of 4 datasets, *logic_reasoning* includes 21 datasets, and *math_reasoning* comprises 16 datasets, among which one is collected by ourselves. All of the above datasets provide original reference answers. In our dataset selection, we do not differentiate between the training and test splits of the original datasets; instead, we only use their questions and reference answers.

# E    LLM Usage For Benchmark Construction

We list all the LLMs we used to generate completions for curated question in Table 7. We carefully selected 22 LLMs from 10 families of both open-source and closed-source models to generate completions. Each model produced one completion for each question, which was collected as candidate completions for subsequent processing.

For all models, we strictly adhered to their licenses and ensured that the generated data would not be used for any model training, but solely for evaluation purposes. For open-source models, we employed the *vLLM* (Kwon et al., 2023) framework for generation. The generation hyper-parameters were set to temperature=0.7 and top_p=0.95, with all other parameters kept at their default values.

Table 7: All LLMs used to generate completions in this paper. We employed multiple open-source and closed-source models of varying scales to generate completions, thereby enhancing the robustness of our constructed benchmark.

| Series | Model |
|---|---|
| OpenAI | gpt-4o-2024-11-20 |
| | gpt-4o-mini |
| anthropic | claude-3.7-sonnet |
| deepseek-math | deepseek-math-7b-instruct (Shao et al., 2024) |
| | deepseek-math-7b-rl (Shao et al., 2024) |
| DeepSeek | DeepSeek-V3 DeepSeek-AI et al. (2025b) |
| | DeepSeek-R1 DeepSeek-AI et al. (2025a) |
| | DeepSeek-R1-Distill-Qwen-7B DeepSeek-AI et al. (2025a) |
| | DeepSeek-R1-Distill-Qwen-32B DeepSeek-AI et al. (2025a) |
| gemma-3 | gemma-3-1b-it (Gemma Team et al., 2025) |
| | gemma-3-4b-it (Gemma Team et al., 2025) |
| | gemma-3-12b-it (Gemma Team et al., 2025) |
| Llama-3 | Llama-3.3-70B-Instruct (Grattafiori et al., 2024) |
| | Llama-3-8B-Instruct (Grattafiori et al., 2024) |
| Qwen2.5 | Qwen2.5-7B-Instruct (Qwen et al., 2025) |
| | Qwen2.5-72B-Instruct (Qwen et al., 2025) |
| Qwen2.5-Math | Qwen2.5-Math-1.5B-Instruct (Yang et al., 2024) |
| | Qwen2.5-Math-7B-Instruct (Yang et al., 2024) |
| | Qwen2.5-Math-72B-Instruct (Yang et al., 2024) |
| QwQ | QwQ-32B (Qwen Team, 2024) |
| Yi-1.5 | Yi-1.5-9B-Chat-16K (AI et al., 2025) |
| | Yi-1.5-34B-Chat (AI et al., 2025) |

# F    Detailed Benchmark Statistics

In this section, we present additional statistics for VerifyBench and VerifyBench-Hard, including the distribution of completion source models for both benchmarks (Appendix F.1) and the distribution of token lengths (Appendix F.2).

## F.1    Source Model Distribution

VerifyBench and VerifyBench-Hard include completions generated by different source models. To better understand their distribution, we report the distribution of source models in both benchmarks, as shown in Table 8.

On VerifyBench and VerifyBench-Hard, the distribution of completions produced by different models does not exhibit extreme imbalance (each model contributes roughly 2–7% on VerifyBench, and 1-12% on VerifyBench-Hard).

Table 8: Distribution of source models which generated the completions in VerifyBench and VerifyBench-Hard.

| Series | Model | VerifyBench | | VerifyBench-Hard | |
|---|---|---|---|---|---|
| | | # of samples | ratio (%) | # of samples | ratio (%) |
| OpenAI | gpt-4o-2024-11-20 | 91 | 4.55 | 0 | 0.00 |
| | gpt-4o-mini | 79 | 3.95 | 0 | 0.00 |
| anthropic | claude-3.7-sonnet | 50 | 2.50 | 0 | 0.00 |
| deepseek-math | deepseek-math-7b-instruct | 107 | 5.35 | 101 | 10.10 |
| | deepseek-math-7b-rl | 92 | 4.60 | 72 | 7.20 |
| DeepSeek | DeepSeek-V3 | 80 | 4.00 | 0 | 0.00 |
| | DeepSeek-R1 | 101 | 5.05 | 0 | 0.00 |
| | DeepSeek-R1-Distill-Qwen-7B | 85 | 4.25 | 0 | 0.00 |
| | DeepSeek-R1-Distill-Qwen-32B | 92 | 4.60 | 112 | 11.20 |
| gemma-3 | gemma-3-1b-it | 148 | 7.40 | 123 | 12.30 |
| | gemma-3-4b-it | 87 | 4.35 | 80 | 8.00 |
| | gemma-3-12b-it | 87 | 4.35 | 7 | 0.70 |
| Llama-3 | Llama-3.3-70B-Instruct | 84 | 4.20 | 0 | 0.00 |
| | Llama-3-8B-Instruct | 117 | 5.85 | 56 | 5.60 |
| Qwen2.5 | Qwen2.5-7B-Instruct | 82 | 4.10 | 106 | 10.60 |
| | Qwen2.5-72B-Instruct | 86 | 4.30 | 18 | 1.80 |
| Qwen2.5-Math | Qwen2.5-Math-1.5B-Instruct | 86 | 4.30 | 49 | 4.90 |
| | Qwen2.5-Math-7B-Instruct | 83 | 4.15 | 91 | 9.10 |
| | Qwen2.5-Math-72B-Instruct | 77 | 3.85 | 21 | 2.10 |
| QwQ | QwQ-32B | 120 | 6.00 | 50 | 5.00 |
| Yi-1.5 | Yi-1.5-9B-Chat-16K | 92 | 4.60 | 21 | 2.10 |
| | Yi-1.5-34B-Chat | 74 | 3.70 | 93 | 9.30 |

## F.2 Token Length Distribution of Completions in VerifyBench(-Hard)

We provide the token-length distributions of completions in both VerifyBench and VerifyBench-Hard, as shown in Figure 4.

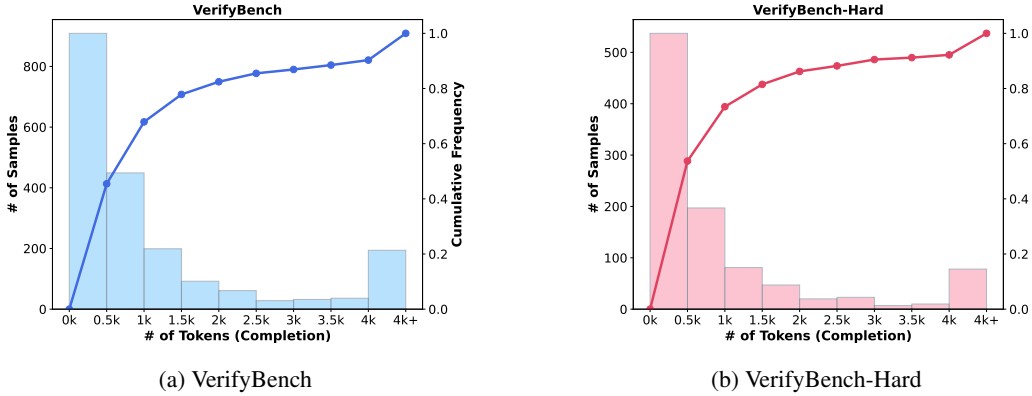

(a) VerifyBench
(b) VerifyBench-Hard

Figure 4: Token length distribution of completions in VerifyBench and VerifyBench-Hard.

It can be seen that the token-length range of the completions in our constructed VerifyBench is broad: it includes shorter chat-style completions as well as longer R1-like completions. This diversity improves the coverage and realism of VerifyBench, making it a more reliable benchmark.

# G    PROMPT TEMPLATES

## G.1    PROMPT TEMPLATE FOR ANSWER TYPE CLASSIFICATION

We present the prompt template we used to generate answer types in Figure 5. {question} and {answer} are placeholders that will be replaced with actual question and answer content.

---

**Prompt for Answer Type Classification**

You are a professional LLM evaluator and now you are tasked with identifying the type of answer that corresponds to a problem, here is what is required of you:
1. I will give you a problem and the corresponding solution, you need to analyze the problem carefully and understand what objective you need to get for that problem;

2. Please choose the most appropriate type for the target (i.e. final answer) to the problem, based on the objective of the give problem, the types available to you include:
A Numeric values
A1 Integers(whole numbers), e.g. 2, 351, etc.
A2 Numerical values containing constants, e.g. 2\pi, etc.
A3 Fractions/Float numbers/Proportions, e.g. \frac{1}{2}, 3.123, 1:2, etc.
A4 Numerical values containing radicals, e.g. \sqrt{3}
A5 Complex numbers, e.g. 3+4i
A6 Angles, e.g. 120^\circ
A7 Non-decimal numbers (not base 10 numeric values), e.g. 1234_8
A8 Multiple values with no order between them, e.g. 1,-2, 1 and 2, etc.
B Mathematical expressions
B1 Algebraic formulas, e.g. ax+b, etc.
B2 Equations or systems of equations, e.g. ax+b=0, etc.
B3 intervals, including concatenation and intersection of intervals, e.g. (1,2] (1,2) \cup (3,4), etc.
B4 A set, a collection of elements, e.g. {1,2,3}, (1,2,3), etc.B5 a matrix or vector, e.g. \begin{pmatrix} -7 \\ 16 \\ 5 \end{pmatrix}, (1, 2) etc.
C Multiple-choice questions
C1 Single-choice questions, choose one correct option from multiple choices
C2 Multiple choice, select multiple correct choices from multiple options
C3 Finite state selection, choose one correct state from a finite number of states, e.g. True/False, yes/no, valid/invalid, etc.
D String
D1 Specific natural language representation, more concerned with whether a particular word or expression is mentioned correctly or not
D2 Semantically consistent natural language representations, where the answer is difficult to be expressed in a single unified word, and there is a preference for comparing the semantic consistency of answers
E Other cases not categorized by the above categories

3. Please analyze carefully and output the serial numbers of the corresponding answer categories, you need to output your reasons for making such a classification. Your output should be in the format of a legal json in {"explanation": "xxx", "type": "xx"}, where 'explanation' is your reasoning, 'type' is the serial number of the final corresponding answer category, and you can output more than one serial number, separated by ','.

4. You only need to care about the type of the final answer.
Question:{question}
Answer:{answer}

---

Figure 5: Prompt for answer type classification. {question} and {answer} are placeholders that will be replaced with actual question and answer content.

## G.2 PROMPT TEMPLATE FOR LLM-AS-A-JUDGE

We present the prompt template we used in LLM-as-a-judge evaluation with a reference answer in Figure 6. {question}, {answer}, and {completion} are placeholders that will be replaced with actual question, answer, and completion content.

---

**Prompt for llm-as-a-judge**

Given the following math problem and the reference answer. Judge the correctness of the answers given later, with some ability to generalize and match the form and format of the answer results. The following specific requirements are followed when judging:

1. Judge only whether the final result of the reference answer and the answer to be judged agree; do not consider whether there are any errors in the process. Don't verify the correctness of the answer by yourself, please only refer to the reference answer for the correctness of the answer.
2. The reference answer and the answer to be judged only need to be essentially the same, ignoring irrelevant details such as units, symbols, whether or not to approximate, and the form of expression in the answer. The two answers are considered to be consistent if they are equivalently transformable.
3. All your analysis answer must be in English.
4. Please analyze the judged answer and try to compare it with the reference answer.
At the end of all analysis, give the result of the judgment on an extra line at the end of the answer in the form "Final Judgment: Yes/No".

Problem: {question}
Reference Answer: {answer}
Solution to be evaluated: {completion}

---

Figure 6: Prompt for LLM-as-a-judge evaluation. {question}, {answer}, and {completion} are placeholders that will be replaced with actual question, answer, and completion content.

## G.3 PROMPT TEMPLATE FOR LLM-AS-A-JUDGE WITHOUT REFERENCE

We present the prompt template we used in LLM-as-a-judge evaluation with a reference answer in Figure 7. {question} and {completion} are placeholders that will be replaced with actual question and completion content.

---

**Prompt for llm-as-a-judge without a Reference Answer**

Given the following math problem, please judge the correctness of the answers given later. The following specific requirements are followed when judging:

1. All your analysis answer must be in English.
2. Please analyze the math problem and the answer and try to tell whether the given completion is a correct answer. At the end of all analysis, give the result of the judgment on an extra line at the end of the answer in the form " Final Judgment: Yes/No ".

Problem: {question}
Solution to be evaluated: {completion}

---

Figure 7: Prompt for LLM-as-a-judge evaluation without reference answers. {question} and {completion} are placeholders that will be replaced with actual question and completion content.

# H DETAILS OF HUMAN ANNOTATION

We present comprehensive details regarding our human annotation process: (1) the composition of our annotation team, including the knowledge backgrounds of annotators, and (2) the annotation interface design (for annotators and for meta-annotators), encompassing annotation instructions and user interface specifications.

## H.1 INSTRUCTION FOR ANNOTATORS

Before initiating the manual annotation process, we conducted a 60-minute video training session for all annotators. After the training, we also provided written instructions detailing the annotation procedure, which were distributed to all annotators. In addition, these instructions were made accessible on the annotation interface itself, enabling annotators to consult them in real time. The following is the instruction text we provided.

> **Instruction for Annotators**
>
> ### What should I do
> **Step 1:** Check the answer type. The page has been pre-annotated, please check whether this annotation matches the actual situation. Please select the most appropriate option from the dropdown list, where option E is not suitable for VerifyBench. If you select option E, you can skip step 2 and directly click the Next Button.
> **Step 2:** Check answer correctness. A total of 4 corresponding answers are displayed, each answer has been pre-annotated for correctness. Please check whether this annotation matches the actual situation, and if it doesn't match, please make the appropriate corrections.
> **Step 3:** Click the Next Button! You have successfully completed the annotation for this case. Thanks for your contribution!

## H.2 ANNOTATORS

Our annotation team consists of 6 members, including 4 annotators and 2 meta-annotators. The meta-annotators are responsible for resolving disagreements that arise during the annotation process.

Among the four annotators, one is an undergraduate student and the other three are master's students, all of whom have engaged in research on LLM reasoning for over three months. The two meta-annotators are current PhD students who have conducted research on LLM reasoning for more than one year. Among the two meta-annotators, one was responsible for resolving disagreements between general reasoning and math reasoning categories, while the other handled conflicts between general reasoning and logic reasoning categories.

Our primary annotator team consists of the following members:

- **Annotator febd:** Undergraduate student with 5 months of experience in LLM reasoning research.
- **Annotator cfaa:** Master's student with 6 months of experience in LLM reasoning research.
- **Annotator ebed:** Master's student with 6 months of experience in LLM reasoning research.
- **Annotator ffce:** Master's degree holder with 1 year of experience in LLM reasoning research.

Our meta-annotator team consists of the following members:

- **Meta-annotator ebed:** Doctoral student with 1 year of experience in LLM reasoning research, specializing in reasoning and logical reasoning domains.
- **Meta-annotator feea:** Doctoral student with 1 year of experience in LLM reasoning research, specializing in reasoning and mathematical reasoning domains.

## H.3 UI DESIGN FOR HUMAN ANNOTATION PROCESS

We developed a web-based annotation interface using Streamlit to facilitate human labeling. All annotations were conducted through the web interface, with different annotators distinguished by unique user parameters. We constructed two distinct interfaces: one for primary annotators and

another for meta annotators. Primary annotators directly labeled the data, while meta annotators resolved disagreements between primary annotators through our double-check verification mechanism and made final labeling decisions based on the annotations from both primary annotators.

### H.3.1 PAGE FOR PRIMARY ANNOTATORS

We designed an annotation interface for annotators that displays the current annotation progress. By clicking on interface elements, annotators can view the complete data in JSON format to make informed judgments. The interface presents one question, one reference answer, and four corresponding completions at a time. Annotators label the answer category using dropdown components and mark the correctness of the four completions using checkboxes. Our annotation interface supports LaTeX formula rendering, enabling annotators to perform mathematical problem annotation in a more readable format. For VerifyBench annotation, the interface displays pre-annotation results, while for VerifyBench-Hard annotation, no pre-annotation results are shown. The interface incorporates an invalid button to enable annotators to quickly mark samples unsuitable for VerifyBench. Any sample marked as invalid by any annotator is excluded from subsequent annotation processes. Additionally, a skip button is implemented to allow annotators to bypass samples they cannot evaluate with confidence. Figure 8 shows the actual annotation interface for primary annotators.

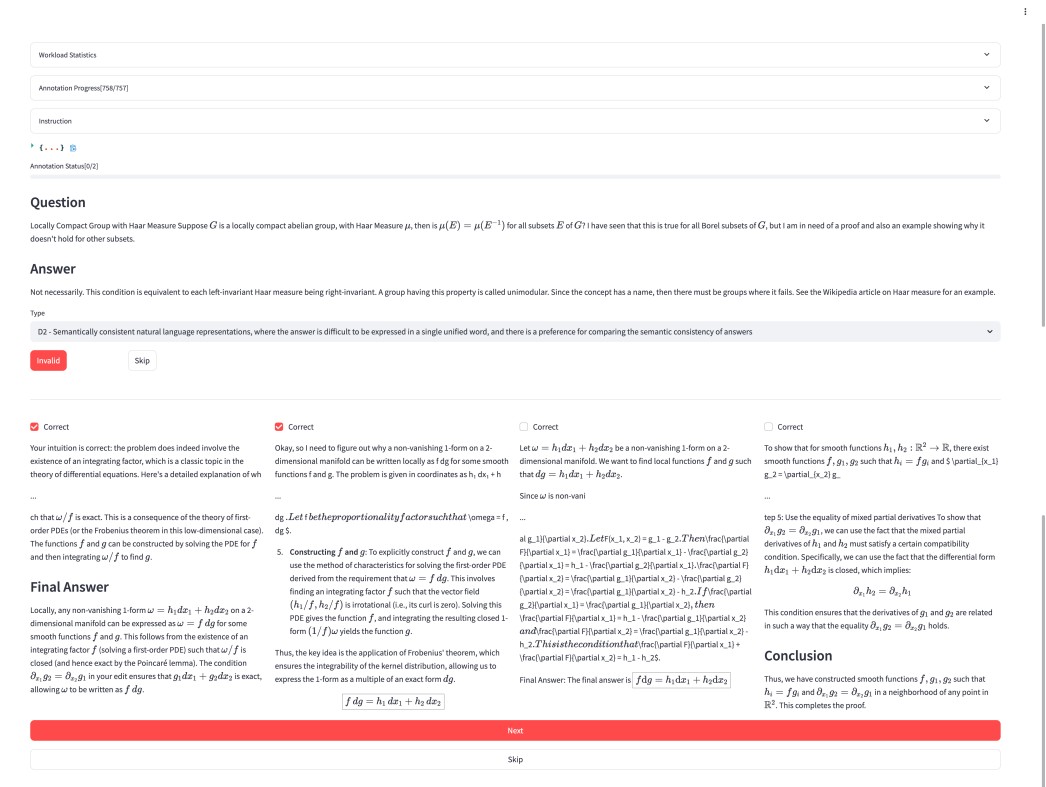

Figure 8: Annotation interface for primary annotators, one sample per page for annotation.

### H.3.2 PAGE FOR META-ANNOTATORS

The interface designed for meta-annotators differs slightly from that of primary annotators. The key distinction lies in the meta-annotators' ability to view detailed annotations from primary annotators, with disagreements between annotators clearly displayed on the page to facilitate final judgment based on the primary annotation information. Furthermore, we designed domain-based filtering element for meta-annotators, enabling them to select domains aligned with their knowledge background and expertise. Other page logic remains identical to the primary annotators' interface. Figure 9 illustrates the actual annotation interface for meta-annotators.

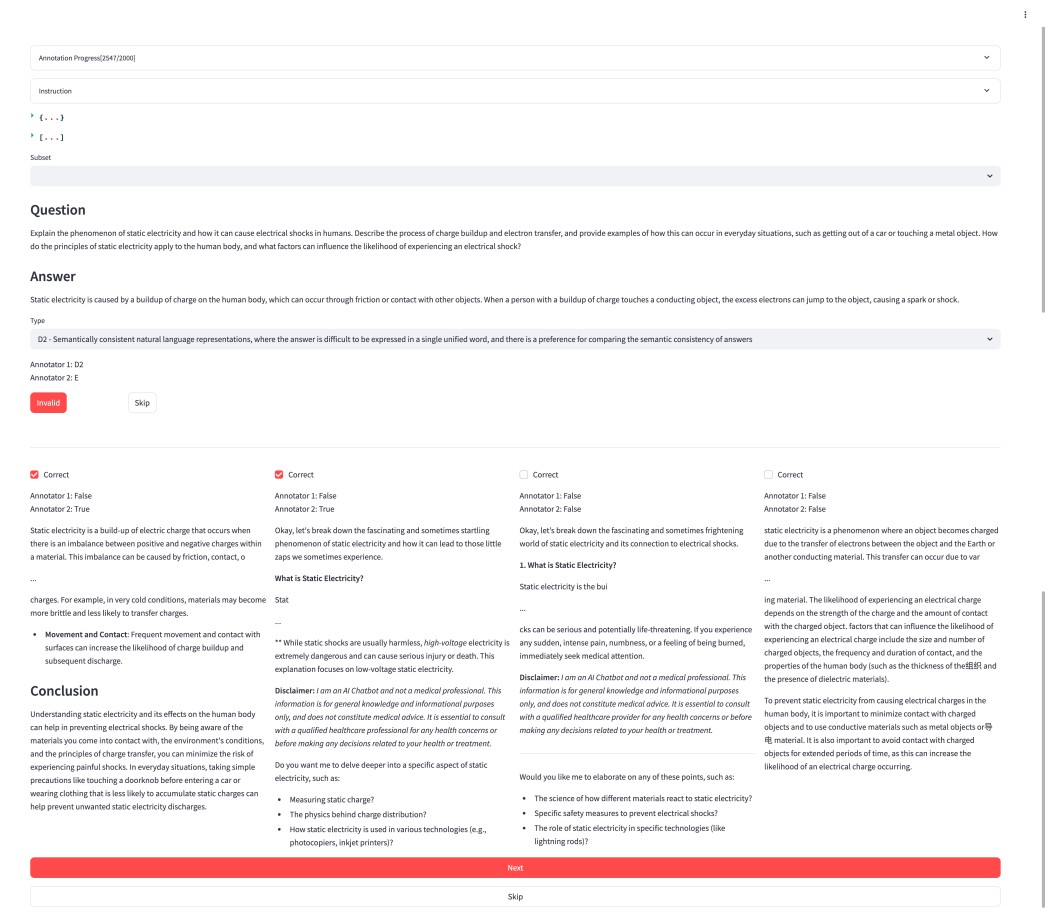

Figure 9: Annotation interface for meta-annotators, one sample per page for annotation.

## I    FULL TAXONOMY OF VERIFYBENCH AND VERIFYBENCH-HARD

During human annotation, we adopted a finer-grained labeling scheme. Beyond the four main categories mentioned in the text (Numeric Values, Expressions, Multi-choice, and String), we further annotated more detailed subcategories within each type for analysis. Specifically, Numeric Values includes 8 subcategories, Expressions includes 5 subcategories, Multi-choice includes 3 subcategories, and String includes 2 subcategories. The descriptions of all subcategories are presented in Table 9.

## J    CORRELATION EXPERIMENTAL DETAILS

Our correlation experiments consist of two parts: we conducted both Reinforcement Learning (RL) (Section J.1) and Rejection Sampling Fine-tuning (RFT) (Section J.2) to validate the usability of VerifyBench.

### J.1    REINFORCEMENT LEARNING EXPERIMENTS

#### J.1.1    SETUP

**Training.**    For RL experiments, we adopted Qwen2.5-3B-Instruct (Qwen et al., 2025) as the base model and conducted training with the DeepMath-103K (He et al., 2025) dataset. For the RL algorithm, we employed GRPO (Shao et al., 2024) optimization, setting the global batch size to 512 and the mini-batch size to 128. For each query, we sampled 16 responses to compute the group relative

Table 9: Full taxonomy and their description of VerifyBench and VerifyBench-Hard

| Label | Type | Sub-Label | Sub-Type Description |
|-------|------|-----------|----------------------|
| A | Numeric Values | A1 | Integers(whole numbers), e.g. 2, 351, etc. |
|   |   | A2 | Numerical values containing constants, e.g. $2\pi$, etc. |
|   |   | A3 | Fractions/Float numbers/Proportions, e.g. $\frac{1}{2}$, 3.123, $1:2$, etc. |
|   |   | A4 | Numerical values containing radicals, e.g. $\sqrt{3}$ |
|   |   | A5 | Complex numbers, e.g. $3 + 4i$ |
|   |   | A6 | Angles, e.g. $120°$ |
|   |   | A7 | Non-decimal numbers (not base 10 numeric values), e.g. $1234_8$ |
|   |   | A8 | Multiple values with no order between them, e.g. $1, -2, 1$ and $2$, etc. |
| B | Expressions | B1 | Algebraic formulas, e.g. $ax + b$, etc. |
|   |   | B2 | Equations or systems of equations, e.g. $ax + b = 0$, etc. |
|   |   | B3 | Intervals, including concatenation and intersection of intervals, e.g. $(1, 2]$ $(1, 2) \cup (3, 4)$, etc. |
|   |   | B4 | A set, a collection of elements, e.g. $\{1, 2, 3\}$, $(1, 2, 3)$, etc. |
|   |   | B5 | A matrix or vector, e.g. $\begin{pmatrix} -7 \\ 16 \\ 5 \end{pmatrix}$, $(1, 2)$ etc. |
| C | Multi-choice | C1 | Single-choice questions, choose one correct option from multiple choices. |
|   |   | C2 | Multiple choice, select multiple correct choices from multiple options. |
|   |   | C3 | Finite state selection, choose one correct state from a finite number of states, e.g. True/False, yes/no, valid/invalid, etc. |
| D | String | D1 | Specific natural language representation, more concerned with whether a particular word or expression is mentioned correctly or not |
|   |   | D2 | Semantically consistent natural language representations, where the answer is difficult to be expressed in a single unified word, and there is a preference for comparing the semantic consistency of answers |
| E | Others | / | Other cases not categorized by the above categories |

advantage. The maximum prompt length was set to 1024, and the maximum response length was set to 7168. For rollouts, we utilized the vLLM (Kwon et al., 2023) engine with temperature fixed at 1.0 and top_p at 1.0. Training was performed with the veRL (Sheng et al., 2025) framework using FSDP as the backend. The learning rate during training was set to 1e-6, with 20 warmup steps. For the LLM verifier, we deployed services through SGLang (Zheng et al., 2024) and performed verification via remote calls. All RL experiments were conducted on 8 H200 GPUs.

**Evaluation.** For evaluation, we leveraged the functionality of the veRL framework to conduct assessments on checkpoints every 10 training steps during RL training. To reduce evaluation variance, we set the temperature to 0.7 and top_p to 0.95, sampling each test example 8 times and then computing the average accuracy.

### J.1.2 EXPERIMENTAL RESULTS

We applied a sliding-window smoothing method to the curves presented in the main text. In Figure 10, we show the original curves before smoothing (light-colored curves).

### J.2 REJECTION SAMPLING FINE-TUNING EXPERIMENTS

### J.2.1 SETUP

**Training.** For the rejection sampling fine-tuning experiments, we used Llama-3.1-8B (Grattafiori et al., 2024) as the base model for SFT. The learning rate was set to a constant value of 1e-5. Training

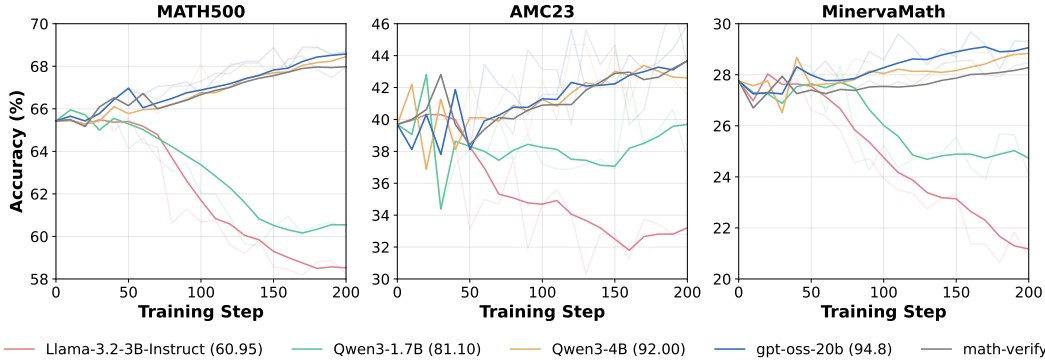

Figure 10: The performance(%) of RL across different LLM judges which have various performance on VerifyBench.

was conducted using the Megatron-LM framework, with a global batch size of 256 and a context length of 4096. To accelerate training, we packed the training samples and trained for one epoch in total. All training experiments were conducted on 32 Ascend H910B-64G GPUs.

**Evaluation.** For evaluation, we used the *vLLM*(Kwon et al., 2023) framework for inference. To reduce evaluation variance, we set the temperature to 0.7 and sampled each test example 16 times, then computed the average accuracy. All inference were conducted on 8 NVIDIA A100-80G.

### J.2.2 EXPERIMENTAL RESULTS

In our experiments, we applied rejection sampling to implement reference-based reward systems. For each question in the GSM8K and MATH training sets, we generated 64 candidate completions using Qwen2.5-Math-7B-Instruct (Yang et al., 2024) with a sampling temperature of 0.7. These responses were subsequently filtered by 4 verifier models with varying performance levels, on VerifyBench: Llama-3.1-8B-Instruct, Qwen3-4B, Qwen3-1.7B and gpt-oss-20b, and a rule-based verifier, math-verify. Only completions consistently verified as correct were retained to form the SFT training data.

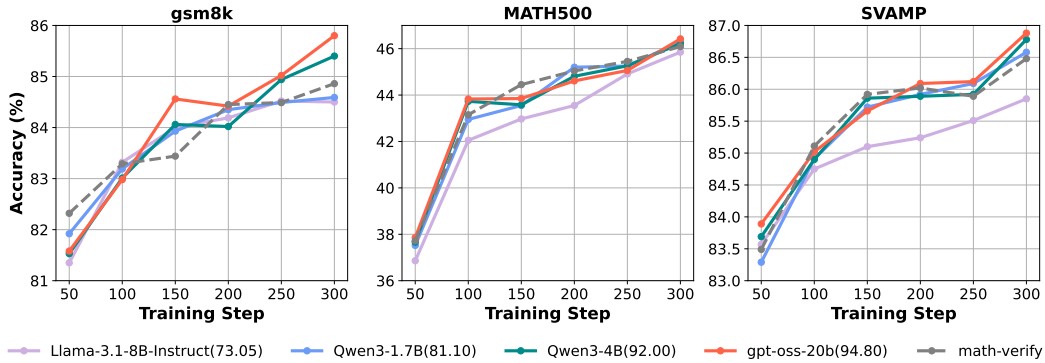

Figure 11: The performance(%) of RFT across different LLM judges and rule-based math-verify which have various performance on VerifyBench.

The resulting models were evaluated on multiple mathematical reasoning benchmarks, including GSM8K (Cobbe et al., 2021), MATH500 (Hendrycks et al., 2021; Lightman et al., 2023), and SVAMP (Patel et al., 2021). As shown in Figure 11, on GSM8K, MATH500, and SVAMP, using gpt-oss-20b, a verifier that achieves higher accuracy on VerifyBench, consistently outperforms Llama-3.1-8B-Instruct, a lower-scoring verifier, under the same training steps. This highlights the strong alignment between VerifyBench and practical usage. Our benchmarks serve as reliable tools to guide the development of reward systems, leading to more effective training and improved model performance.

## K EXAMPLES OF VERIFYBENCH

In this section, we provide representative samples from the four answer-type categories in Verify-Bench to facilitate a clearer understanding of the benchmark we constructed. Due to space limitations, we omitted part of the text and used `<omitted>` as a placeholder.

Figure K.1 presents a sample from the **Numeric Values** category, Figure K.2 shows a sample from the **Expression** category, Figure K.3 illustrates a sample from the **Multi-choice** category, and Figure K.4 displays a sample from the **String** category.

### K.1 AN EXAMPLE OF VERIFYBENCH (ANSWER TYPE: NUMERIC VALUES)

---

**VerifyBench (Answer type: Numeric Values)**

**Question:**

```
How many positive and negative integers is $12$ a multiple of?
```

**Reference answer:**

```
The number $12$ is a multiple of $-12, -6, -4, -3, -2, -1, 1,
2, 3, 4, 6,$ and $12,$ for a total of $\boxed{12}$ integers.
```

**Completion:**

```
We want to find the number of positive and negative integers
that divide $12$.
The positive integers that divide $12$ are the positive
divisors of $12$. The divisors of $12$ are the integers that
divide $12$ without leaving a remainder.
We can find the divisors of $12$ by listing pairs of integers
that multiply to $12$.
$1 \times 12 = 12$
$2 \times 6 = 12$

<omitted>

The number of positive integers that divide $12$ is 6.
The number of negative integers that divide $12$ is 6.

Final Answer: The final answer is $\boxed{6}$
```

**Golden Label:** ❌ **False**

---

### K.2 AN EXAMPLE OF VERIFYBENCH (ANSWER TYPE: EXPRESSION)

---

**VerifyBench (Answer type: Expression)**

**Question:**

```
Consider two lines parameterized by:
\[
\begin{pmatrix} x \\ y \end{pmatrix} = \begin{pmatrix} 1 \\ 2
\end{pmatrix} + t \begin{pmatrix} 2 \\ -3 \end{pmatrix}\n\]\
nand\n\[\n\begin{pmatrix} x \\ y \end{pmatrix} = \begin{
pmatrix} 4 \\ 5 \end{pmatrix} + u \begin{pmatrix} 1 \\ -1 \end
{pmatrix}
\]
```

```
Determine the coordinates of their intersection point.
```

**Reference answer:**

```
For the first line, \[ \begin{pmatrix} x \\ y \end{pmatrix} =
\begin{pmatrix} 1 \\ 2 \end{pmatrix} + t \begin{pmatrix} 2 \\
-3 \end{pmatrix} = \begin{pmatrix} 1 + 2t \\ 2 - 3t \end{
pmatrix} \] For the second line,

<omitted>

2 - 3(-6) \end{pmatrix} = \begin{pmatrix} 1 - 12 \\ 2 + 18 \
end{pmatrix} = \begin{pmatrix} -11 \\ 20 \end{pmatrix} \]
So, the intersection point is \( \boxed{\begin{pmatrix} -11 \\
 20 \end{pmatrix}} \).
```

**Completion:**

```
Okay, so I have to find the intersection point of two lines
given in parametric form. The first line is parameterized as:

<omitted>

Thus, the coordinates of the intersection point are \(\boxed
{(-11, 20)}\).
```

**Golden Label:** ✅ True

## K.3   AN EXAMPLE OF VERIFYBENCH (ANSWER TYPE: MULTI-CHOICE)

**VerifyBench (Answer type: Multi-choice)**

**Question:**

```
Given that point $P$ lies on the circle $C: x^{2}-6x+y^{2}-6y
+14=0$, and the line $AB: x+y-2=0$, then which of the
following statements is true?
A: Line $AB$ intersects circle $C$.
B: Line $AB$ is disjoint from circle $C$.
C: The distance from point $P$ to line $AB$ is greater than $\
frac{1}{2}$.
D: The distance from point $P$ to line $AB$ is less than $5$.
```

**Reference answer:**

```
To analyze the relationship between circle $C$ and line $AB$,
we first need to rewrite the equation

<Omitted>

And since the maximum distance is $2\sqrt{2} + 2$, which is
less than $5$, statement D is also true. Therefore, the
correct choices are $\boxed{BCD}$.
```

**Completion:**

```
To analyze the problem, let's break it down step by step.
---
```

```
### Step 1: Rewrite the equation of the circle \( C \) The
given equation of the circle is: \[ x^2 - 6x + y^2 - 6y + 14 =
 0. \] Complete the square for both \( x \)

<omitted>

which is: \[ 2\sqrt{2} - 2. \] Numerically: \[ 2\sqrt{2} \
approx 2.828, \quad 2\sqrt{2} - 2 \approx 0.828, \quad 2\sqrt
{2} + 2 \approx 4.828. \] Thus, the distance from \( P \) to
the line \( AB \) is always greater than \( \frac{1}{2} \) and
 less than \( 5 \). Therefore, both **statement C** and **
statement D** are true.

---

### Final Answer: The true statements are:
- **B**: Line \( AB \) is disjoint from circle \( C \).
- **C**: The distance from point \( P \) to line \( AB \) is
greater than \( \frac{1}{2} \).
- **D**: The distance from point \( P \) to line \( AB \) is
less than \( 5 \).
```

**Golden Label:** ✅ True

## K.4 AN EXAMPLE OF VERIFYBENCH (ANSWER TYPE: STRING)

**VerifyBench (Answer type: String)**

**Question:**

```
Context:
Brian is a swan.
Bernhard is a frog.
Lily is a rhino.
Bernhard is white.
Greg is a swan.
Brian is gray.
Greg is gray.
Lily is yellow.
Julius is a frog.

What color is Julius?
```

**Reference answer:**

```
white
```

**Completion:**

```
The color of Julius is not explicitly mentioned in the
provided context. Therefore, we cannot determine what color
Julius is based on the information given.
```

**Golden Label:** ❌ False

## L  EXAMPLES OF VERIFYBENCH-HARD

In this section, we present hard-to-verify samples from the four different answer-type categories in VerifyBench-Hard, enabling a clearer understanding of the construction and characteristics of VerifyBench-Hard. Due to space limitations, we omitted part of the text and used `<omitted>` as a placeholder.

Figure L.1 presents a sample from the **Numeric Values** category, Figure L.2 shows a sample from the **Expression** category, Figure L.3 illustrates a sample from the **Multi-choice** category, and Figure L.4 displays a sample from the **String** category.

### L.1  AN EXAMPLE OF VERIFYBENCH-HARD (ANSWER TYPE: NUMERIC VALUES)

---

**VerifyBench-Hard (Answer type: Numeric Values)**

**Question:**

```
Given the function $f(x) = \cos x - \sin x$, where $f''(x)$ is
 the second derivative of the function $f(x)$, find the value
of $f''(\frac{\pi}{6})$.
```

**Reference answer:**

```
First, let's find the first and second derivatives of the
function $f(x)$.

<ommited>

Rearrange the fraction to match one of the answer choices:

$$= \boxed{-\frac{1 - \sqrt{3}}{2}}$$
```

**Completion:**

```
To find the value of \( f''\left(\frac{\pi}{6}\right) \) for
the function \( f(x) = \cos x - \sin x \), we need to follow
these steps:

1. **Find the first derivative \( f'(x) \):**

The function \( f(x) \) is given by:
\[
f(x) = \cos x - \sin x
\]

<ommited>

Thus, the value of \( f''\left(\frac{\pi}{6}\right) \) is:
\[
\boxed{\frac{1 - \sqrt{3}}{2}}
\]
```

**Golden Label:** ❌ **False**

---

## L.2 AN EXAMPLE OF VERIFYBENCH-HARD (ANSWER TYPE: EXPRESSION)

**VerifyBench-Hard (Answer type: Expression)**

**Question:**

```
$A$ and $B$ are constants such that
$$\frac{4x+5}{x^2+x-2}= \frac{A}{x+2} +\frac{B}{x-1}. $$Find
$$\frac{B}{x+1} - \frac{A}{x-2}.$$
```

**Reference answer:**

```
We have that
$$\frac{4x+5}{x^2+x-2}= \frac{4x+5}{(x+1)(x-2)}=\frac{A}{x+2}
+\frac{B}{x-1}. $$Multiplying both sides by $(x+1)(x-2)$ gives
$$4x+5=A(x+1)+B(x-2)$$which we can expand to get
$$4x+5=(A+B)x+A-2B.$$By comparing coefficients, we see that $A
+B=4$ and $A-2B=5$. We can solve these equations to get $A=1$
and $B=3$.
Then
$$\begin{aligned} \frac{B}{x+1} - \frac{A}{x-2}&=\frac{3}{x+1}
 - \frac{1}{x-2}\\
&=\frac{3(x-2)-1(x+1)}{(x+1)(x-2)}\\
&=\boxed{\frac{2x-7}{(x+1)(x-2)}}.
\end{aligned}$$
```

**Completion:**

```
To solve the given problem, we start by expressing the right-
hand side of the equation with a common denominator:

<ommited>

This expression is already in its simplest form, so the final
answer is:

\[
\boxed{\frac{3}{x+1} - \frac{1}{x-2}}.
\]
```

**Golden Label:** ✅ **True**

## L.3 AN EXAMPLE OF VERIFYBENCH-HARD (ANSWER TYPE: MULTI-CHOICE)

**VerifyBench-Hard (Answer type: Multi-choice)**

**Question:**

```
Context:
In the criminal justice system, investigators aim to solve
crimes. They knew solid detective work led to clues, and
forensic analysis provided evidence. However, some crimes went
 unsolved, lacking clues or evidence. They knew if there were
sufficient resources, detection would be thorough.
Question:
The forensics team found evidence. From the context, did the
investigation team have sufficient resources?
yes or no?
```

**Reference answer:**

```
no
```

**Completion:**

```
The context does not explicitly state whether the
investigation team had sufficient resources or not. It only
mentions that solid detective work and forensic analysis are
involved in solving crimes and that some crimes go unsolved
due to lack of clues or evidence. So, the answer is:

No, the context does not provide enough information to
determine if the investigation team had sufficient resources.
```

**Golden Label:** ✅ True

## L.4 AN EXAMPLE OF VERIFYBENCH-HARD (ANSWER TYPE: STRING)

**VerifyBench-Hard (Answer type: String)**

**Question:**

```
Sort the following words alphabetically: List: wintertime
mobster ugh brown fleabane longish ipsilateral cortex hair
diminutive trapezoidal guy richfield highfalutin borough foot
delphinium cosec abstract
```

**Reference answer:**

```
abstract borough brown cortex cosec delphinium diminutive
fleabane foot guy hair highfalutin ipsilateral longish mobster
 richfield trapezoidal ugh wintertime
```

**Completion:**

```
Here's the list of words sorted alphabetically:

1. Abstract
2. borough
3. Brown
4. Cosec
5. Delphinium
6. Diminutive
7. Fleabane
8. Guy
9. Hair
10. Highfalutin
11. ipsilateral
12. Longish
13. Mobster
14. Richfield
15. Trapezoidal
16. Ugh
17. Wintertime
```

**Golden Label:** ❌ **False**

## M  ERROR CASE STUDY

To facilitate a deeper understanding of the error patterns exhibited by current models on Verify-Bench, we present four representative failure cases of gpt-oss-20b. These examples span multiple answer types, illustrating the breadth of scenarios in which the model's verification judgments break down.

**Case 1 (Answer type: Numeric Values)**

**Question:**

```
Zoe picked five apples from her tree. Now the tree has six
apples still on it. What's the difference between the number
of apples Zoe picked and now the tree has?
```

**Reference answer:**

```
1
```

**Completion:**

```
To solve the problem, we need to determine the difference
between the number of apples Zoe picked and the number of
apples still on the tree.

First, let's identify the number of apples Zoe picked.
According to the problem, Zoe picked 5 apples.

Next, we need to determine the number of apples still on the
tree. The problem states that after Zoe picked the apples, the
 tree still has 6 apples.

Now, we calculate the difference between the number of apples
Zoe picked and the number of apples still on the tree. This
can be done by subtracting the number of apples still on the
tree from the number of apples Zoe picked:

\[ 5 - 6 = -1 \]

Therefore, the difference between the number of apples Zoe
picked and the number of apples still on the tree is \(\boxed
{-1}\).
```

**Judgment:**

```
We need to judge if the evaluated answer's final result
matches reference answer. Reference answer: 1. Evaluated
answer gives -1. Are they considered essentially same? No, 1
vs -1 not equivalent. So mismatch. Therefore judgment No.
The evaluated solution computes the difference as \(5-6 = -1\)
, giving a final result of \(-1\). The reference answer states
 the correct result is \(1\). Since \(-1\) and \(1\) are not
equivalent (they differ in sign), the evaluated answer does
not match the reference answer.

**Final Judgment: No**
```

**Golden Label:** ✅ **True**

---

**Case 2 (Answer type: Expression)**

**Question:**

```
A line passing through the left focus $F_1$ of a hyperbola and
 perpendicular to the real axis of the hyperbola intersects
the hyperbola at points $A$ and $B$. If there exists a point
$C$ on the line of the imaginary axis of the hyperbola such
that $\vec{AC} \cdot \vec{BC} = 0$, then the range of the
eccentricity $e$ of the hyperbola is _______.
```

**Reference answer:**

```
Let the equation of the hyperbola be $\frac{x^2}{a^2} - \frac{
y^2}{b^2} = 1$, with $A\left(-c, \frac{b^2}{a}\right)$, $B\
left(-c, -\frac{b^2}{a}\right)$, and $C(0, t)$.

From $\vec{AC} \cdot \vec{BC} = 0$,
```

```
we get $t^2 = \frac{b^4}{a^2} - c^2 \geqslant 0$, and $e \
geqslant \frac{\sqrt{5} + 1}{2}$.

Therefore, the range of the eccentricity $e$ of the hyperbola
is $\boxed{e \geqslant \frac{\sqrt{5} + 1}{2}}$.
```

**Completion:**

```
To solve the problem, we need to analyze the given conditions
and use the properties of the hyperbola. Let's start by
defining the hyperbola and its properties.

<ommited>

Therefore, the range of the eccentricity \( e \) is:
\[ \boxed{\left[ \sqrt{\frac{3 + \sqrt{5}}{2}}, \infty \right)
}. \]
```

**Judgment:**

```
We need to compare final result with reference: reference says
 e \ge (sqrt5+1)/2 \approx (2.236+1)/2=1.618. The evaluated
answer says e \ge sqrt((3+sqrt5)/2). Compute: sqrt5\approx2
.236, 3+2.236=5.236, /2=2.618, sqrt\approx1.618. Indeed sqrt
((3+sqrt5)/2) equals (sqrt5+1)/2? Let's check algebra: (sqrt5
+1)/2 squared = (5+2sqrt5+1)/4 = (6+2sqrt5)/4 = (3+sqrt5)/2.
So sqrt((3+sqrt5)/2) = (sqrt5+1)/2 (positive). So the
evaluated answer's range is e \ge sqrt((3+sqrt5)/2) which
equals (sqrt5+1)/2. So they match. The answer also gave
interval [that, \infty). Reference gave e \ge that. So
consistent. Therefore final judgment Yes.
The evaluated answer ultimately states that the eccentricity
must satisfy

\[
e \ge \sqrt{\frac{3+\sqrt5}{2}},
\]

and notes that this is the lower bound for \(e\).  Using the
identity

\[
\sqrt{\frac{3+\sqrt5}{2}}=\frac{\sqrt5+1}{2},
\]

the lower bound is exactly the same as the one given in the
reference answer, which is \(e\ge\frac{\sqrt5+1}{2}\). Both
answers therefore give the same range for the eccentricity (
the evaluated answer additionally writes the range as \([\,\
frac{\sqrt5+1}{2},\infty)\), which is equivalent to the
reference's \(e\ge\frac{\sqrt5+1}{2}\)).

**Final Judgment: Yes**
```

**Golden Label:** ❌ False

**Case 3 (Answer type: Multi-choice)**

**Question:**

```
Given the function $f(x) = 2^{2x} − 2^{x+1} + 2$, with its
domain as $M$ and its range as $[1, 2]$, the correct statement
(s) among the following is/are
(1) $M = [0, 2]$; (2) $M = (−\infty, 1]$; (3) $M \subseteq (−\
infty, 1]$; (4) $0 \in M$; (5) $1 \in M$.
```

**Reference answer:**

```
Since $f(x) = 2^{2x} − 2^{x+1} + 2 = (2^x − 1)^2 + 1 \in [1,
2]$,
$\therefore 2^x − 1 \in [−1, 1]$, which means $2^x \in [0, 2]$
$\therefore x \in (−\infty, 1]$, which means the domain of the
 function $f(x) = 2^{2x} − 2^{x+1} + 2$ is $(−\infty, 1]$.
When the function's minimum value is 1, it only happens at $x
=0$,
thus, (4) $0 \in M$ is correct,
When the function value is 2, it only happens at $x=1$, thus,
(5) $1 \in M$ is correct
Also, it must be that $M \subseteq (−\infty, 1]$; hence, (3)
is correct
When $M = [0, 1]$, at this time, the range of the function is
$[1, 2]$, $\therefore M = (−\infty, 1]$ is not necessarily
correct, hence, (2) is incorrect;
When $x=2$, the function value is 10, hence, (1) $M = [0, 2]$
is incorrect.
In summary, the correct statement numbers are $\boxed{(3), (4)
, (5)}$.
```

**Completion:**

```
We are tasked with analyzing the function \( f(x) = 2^{2x} −
2^{x+1} + 2 \), its domain \( M \), and its range \( [1, 2] \)
. Let's carefully evaluate the given statements.

<ommited>

Now we evaluate the given statements:

1. \( M = [0, 2] \): **False**, since \( M = [0, 1] \), not \(
 [0, 2] \).
2. \( M = (−\infty, 1] \): **False**, since \( M = [0, 1] \),
not \( (−\infty, 1] \).
3. \( M \subseteq (−\infty, 1] \): **True**,
```

**Judgment:**

```
We need to judge the solution to be evaluated (the "Solution
to be evaluated") against the reference answer. The reference
answer says the correct statements are (3), (4), (5). The
solution to be evaluated says: It goes through steps, and at
the end it says: "Thus, the domain M is [0, 1]." Then it says:
 "Now we evaluate the given statements: 1. M = [0, 2]: False,
2. M = (−\infty, 1]: False, 3. M \subseteq (−\infty, 1]: True,
 4. 0 \in M: True, 5. 1 \in M: True." So the solution to be
evaluated says the correct statements are (3), (4), (5). That
```

```
matches the reference answer. So the final judgment is Yes. We
 need to output the final judgment line.
The solution to be evaluated concludes that the correct
statements are (3), (4), and (5), which matches the reference
answer's conclusion that the correct statements are (3), (4),
and (5). Therefore, the answers agree.

Final Judgment: Yes
```

**Golden Label:** ❌ **False**

---

**Case 4 (Answer type: String)**

**Question:**

```
Context:
Greg is a lion.
Julius is a swan.
Lily is a lion.
Greg is yellow.
Bernhard is a lion.
Brian is a lion.
Brian is yellow.
Julius is white.
Bernhard is yellow.

What color is Lily?
```

**Reference answer:**

```
yellow
```

**Completion:**

```
According to the context, Lily is a lion, and there is no
information about her color. Since Greg, Bernhard, and Brian
are all lions and are yellow, it is possible that Lily is also
 yellow, but it is not explicitly stated.
```

**Judgment:**

```
We need to judge if the solution to be evaluated agrees with
reference answer. Reference answer: "yellow". Evaluated
solution says: "it is possible that Lily is also yellow, but
it is not explicitly stated." This does not assert yellow
definitively. The final result is not "yellow". So does it
agree? The requirement: judge only whether final result of
reference answer and answer to be judged agree. The evaluated
answer's final result is ambiguous, not a definitive color. So
 does that match "yellow"? No. Therefore judgment: No.
The evaluated answer does not definitively state that Lily's
color is yellow; it only suggests that yellow is possible and
notes the lack of explicit information. The reference answer
definitively states "yellow." Because the evaluated answer's
final result is not the same definitive color, it does not
agree with the reference answer.

Final Judgment: No
```

**Golden Label:** ✅ True

