# OpenReview forum: "VerifyBench: Benchmarking Reference-based Reward Systems for Large Language Models"
_ICLR.cc/2026/Conference — ICLR 2026 Poster_

### Official Review · Reviewer_uUti · 2025-10-17

**Soundness:** 4
**Presentation:** 4
**Contribution:** 3
**Rating:** 8
**Confidence:** 4

**Summary:**

This paper introduces VerifyBench and VerifyBench-Hard, two benchmarks for evaluating reference-based reward systems in reinforcement learning for large language model reasoning. Unlike previous reward benchmarks that rely on pairwise preference judgments, VerifyBench assesses whether the reward models are able to identify if a model-generated response is correct or not, given reference answers. The datasets are built through large-scale curation and careful human annotation.

**Strengths:**

- The work is timely, filling a clear gap in the evaluation landscape as the community shifts from pairwise preference to reference-based reward evaluation in RL for reasoning models.

- The pipeline ensures high diversity (spanning general, logical, and mathematical reasoning), and the use of human annotators for verification improves reliability and validity.

- The RL correlation experiment convincingly demonstrates that verifier quality (as measured by VerifyBench) directly impacts downstream RL performance, emphasizing the benchmark’s practical utility.

**Weaknesses:**

- Despite the diversity of question types, coding-related tasks (e.g., code generation, debugging) are notably absent.

- The benchmark does not include regex-based verifiers, which are widely used in practice.

**Questions:**

- Could you clarify the difference between LLM-as-a-judge and Model-based Verifier? Since both involve LLMs producing binary verification signals, what distinctions make the latter uniquely "model-based"?

- In the RL correlation results, the VerifyBench-Hard scores suggest that stronger verifiers (e.g., gpt-oss-20b) outperform weaker ones (e.g., Qwen3-4B), yet this does not translate linearly into downstream RL gains (Figure 3). What do you think are the sources of this mismatch?

- Do you plan to maintain a public and continuously updated leaderboard (similar to RewardBench) to track verifier progress over time? Such a platform could help make VerifyBench the de facto standard for reference-based reward evaluation.

---

> ### Author Response · Authors · 2025-11-23
> **Rebuttal [1/2]**
>
> Dear Reviewer ApHj,
>
> Thank you for the effort and time you dedicated to reviewing our paper, and for your recognition the contribution of our work. We have greatly benefited from your valuable feedback. Below are our responses to the weaknesses and questions you pointed out:
>
> ---
>
> > **[Weakness 1]: Despite the diversity of question types, coding-related tasks (e.g., code generation, debugging) are notably absent.**
>
> Thank you for your question. When we initially designed the domains of VerifyBench, we also considered whether code reasoning should be included as a target domain. However, after extensive investigation, we found that **existing reasoning models rarely rely on model-based verifiers during training on coding-related tasks[1-2]**. Instead, they primarily use carefully constructed test cases combined with large-scale sandbox environments for verification.
>
> We believe that code reasoning tasks seldom introduce substantial ambiguity during evaluation, as both the extraction and verification of answers are relatively straightforward. Fixed test cases are typically sufficient for reliable verification. In contrast, model-based verifiers excel in scenarios where answers are difficult to extract or can appear in multiple valid forms. For this reason, we did not include code reasoning as a target domain.
>
> Ref:
> [1] Kimi Team, Kimi K2: Open Agentic Intelligence.
> [2] MiniMax Team, MiniMax-M1: Scaling Test-Time Compute Efficiently with Lightning Attention.
>
>
> > **[Weakness 2]: The benchmark does not include regex-based verifiers, which are widely used in practice.**
>
> Thank you for your question. In the main evaluation results reported in Table 2, we have already included a widely used regex-based verifier, *math-verify*. This verifier performs reasonably well on math-related tasks such as those involving numbers and expressions, but its performance degrades noticeably on multi-choice and string categories.

---

> > ### Author Response · Authors · 2025-11-23
> > **Rebuttal [2/2]**
> >
> > > **[Question 1]: Could you clarify the difference between LLM-as-a-judge and Model-based Verifier? Since both involve LLMs producing binary verification signals, what distinctions make the latter uniquely "model-based"?**
> >
> > Thank you for your question. In Table 1, we compare two categories of LLM-based verifiers, **LLM-as-a-judge** and **Model-based Verifier**, and summarize their key differences below:
> >
> > * **LLM-as-a-judge (training-free):** This paradigm directly employs a general-purpose instruct LLM to evaluate a given sample using a verification prompt, requiring the model to produce judgments in a predefined format. Since it does not involve any task-specific training, it relies entirely on the inherent verification capability of off-the-shelf LLMs.
> > * **Model-based Verifier (training-based):** This approach explicitly trains an existing model on verification tasks, enabling it to achieve stronger verification accuracy. It requires constructing a dedicated verification dataset and performing fine-tuning (e.g., via SFT or RL) to specialize the model for verification.
> >
> > > **[Question 2]: In the RL correlation results, the VerifyBench-Hard scores suggest that stronger verifiers (e.g., gpt-oss-20b) outperform weaker ones (e.g., Qwen3-4B), yet this does not translate linearly into downstream RL gains (Figure 3). What do you think are the sources of this mismatch?**
> >
> > Thank you for raising this important question. We argue that **the effectiveness of a verifier within RL training is strongly conditioned on the characteristics of the RL data itself**. When RL is conducted on data that is intrinsically easy to verify, such as problems whose golden answers are integers [1], relatively weak verifiers are sufficient, as they already achieve high accuracy on such constrained tasks. In contrast, building more capable and general-purpose LLMs requires RL on data that is substantially more diverse and heterogeneous. This is why many state-of-the-art models increasingly rely on carefully trained model-based verifiers to handle complex and ambiguous verification scenarios[2-3].
> >
> > To assess verification performance under such challenging conditions, we introduce VerifyBench-Hard, which targets samples that are significantly more complex and hard to verify. Notably, **the samples in VerifyBench-Hard are much more hard-to-verify than those used in our RL correlation experiments**. Although gpt-oss-20b surpasses Qwen3-4B on VerifyBench-Hard, the two models perform similarly on VerifyBench, which explains the observed discrepancy. The RL correlation experiments utilize DeepMath-103K, a curated mathematics dataset containing relatively easy-to-verify problems. We anticipate that once RL training begins to incorporate more general, complex, and diverse datasets, currently absent from the open-source landscape, the discriminative power of VerifyBench-Hard will become substantially more pronounced.
> >
> > Ref:
> > [1] Yu et al. DAPO: An Open-Source LLM Reinforcement Learning System at Scale.
> > [2] Qwen Team, Qwen3 Technical Report.
> > [3] Meituan LongCat Team, Introducing LongCat-Flash-Thinking: A Technical Report.
> >
> > > **[Question 3]: Do you plan to maintain a public and continuously updated leaderboard (similar to RewardBench) to track verifier progress over time? Such a platform could help make VerifyBench the de facto standard for reference-based reward evaluation.**
> >
> > Yes, we are following the approach of RewardBench to maintain a leaderboard on Hugging Face. We are actively working on building this Space and hope to make it publicly available soon.
> >
> > ---
> >
> > Thank you again for the thorough review. We think that our rebuttal can address your main concern, if you have any further questions, we would be happy to answer them.
> >
> > Sincerely,
> > VerifyBench Authors

---

> > > ### Comment · Reviewer_uUti · 2025-11-25
> > >
> > > Thank you for your reply to my questions.
> > > Personally, I find the distinction between LLM-as-a-judge and Model-based Verifier a bit artificial, since verification datasets are included in all large-scale model training, as the kind you use in LLM-as-a-judge.
> > > However, I still think this is a good work with interest for the community, and as such I keep my recommendation to accept the paper.

---

> ### Author Response · Authors · 2025-11-25
>
> Thank you for your response and recommendation!
>
> We would like to take this opportunity to further clarify the distinction between LLM-as-a-judge and model-based verifiers. We view these as two different categories of LLMs. LLM-as-a-judge refers to general-purpose LLMs, whereas model-based verifiers are verification-specialized LLMs. This distinction is analogous to the relationship between the Qwen2.5 series and the Qwen2.5-Math series. **Model-based verifiers are explicitly optimized for verification tasks and typically achieve superior verification performance compared to general-purpose models of the same size, though this specialization often comes at the cost of reduced capability on other tasks [1-2].**
>
> Ref:
> [1] Chen, et al. xVerify: Efficient Answer Verifier for Reasoning Model Evaluations.
> [2] Liu, et al. CompassVerifier: A Unified and Robust Verifier for LLMs Evaluation and Outcome Reward
>
> ---
>
> Once again, we sincerely appreciate the time and effort you devoted during the review process. Your feedback has been invaluable in improving our paper. We look forward to future opportunities for further discussion.😄

---

### Official Review · Reviewer_ApHj · 2025-10-25

**Soundness:** 3
**Presentation:** 3
**Contribution:** 3
**Rating:** 6
**Confidence:** 4

**Summary:**

This paper introduces VerifyBench and its more challenging variant, VerifyBench-Hard, as two new benchmarks designed for the systematic evaluation of Reference-Based Verifiers. The two benchmarks cover four distinct answer formats and three different reasoning task types. Through extensive experimentation involving Rule-Based Verifiers, LLM-as-a-Judge methods, and Model-Based Verifiers, this study reveals that while current state-of-the-art model-based approaches perform well in general testing, VerifyBench-Hard still poses a significant challenge to them. The paper further explores the performance of traditional ORMs and experimentally confirms that a higher score on VerifyBench positively correlates with the greater performance improvement the verifier provides during RL training.

**Strengths:**

1. The benchmark provides a validated evaluation framework that enables quantitative assessment of verifiers.

2. The test setting is diverse and challenging, and it is able to distinguish even among SOTA models.

3. The experiments empirically demonstrate a correlation between higher benchmark scores and better training performance.

**Weaknesses:**

1. The paper does not appear to include examples of difficult samples. Showing more complex and challenging instances from the benchmark would be convicing.

2. The benchmark is mainly focused on mathematics, so it may not reflect performance on verifiable tasks across the broader domain, such as evaluating physics expressions or chemical equations.

**Questions:**

1.RL training with Qwen-4B as the verifier clearly outperforms training with weaker models, while OSS does not yield additional gains. Does this suggest that there is in fact some “threshold” that determines whether a verifier is basically reasonable, and that once this threshold is reached, further gains become minor?

2.In the RFT experiments, were models stronger than Qwen-4B (e.g., OSS used in the RL experiments) also evaluated? If so, how did they perform?

3.Can you provide some challenging examples from VerifyBench and illustrate how the models judged them correctly or incorrectly? In particular, for the OSS model, examining its reasoning traces may clarify why it makes incorrect decisions. This could in turn inform future verifier improvements.

---

> ### Author Response · Authors · 2025-11-23
> **Rebuttal [1/2]**
>
> Dear Reviewer ApHj,
>
> Thank you for the effort and time you dedicated to reviewing our paper, and for your recognition of our benchmarks and related experiments. We have greatly benefited from your valuable feedback. Below are our responses to the weaknesses and questions you pointed out:
>
> ---
>
> > **[Weakness 1]: The paper does not appear to include examples of difficult samples. Showing more complex and challenging instances from the benchmark would be convicing.**
>
> Thank you for the constructive issue. In the new revision, we have added more examples in Appendix J and Appendix K, including 4 examples from VerifyBench and 4 from VerifyBench-Hard that cover different answer types. These examples allow readers to better understand the benchmark we constructed.
>
> > **[Weakness 2]: The benchmark is mainly focused on mathematics, so it may not reflect performance on verifiable tasks across the broader domain, such as evaluating physics expressions or chemical equations.**
>
> Thank you for your question. In developing VerifyBench and VerifyBench-Hard, we curated data from three major domains: **math reasoning**, **logic reasoning**, and **general reasoning**. The *general reasoning* category spans a wider range of scientific and commonsense reasoning tasks, including physics and chemistry, but its relative proportion in the current benchmarks is indeed smaller.
>
> VerifyBench includes four distinct answer types. Based on our analysis of existing open-source datasets, the **math reasoning domain provides broad coverage across all answer types**, which explains why it contributes a larger number of samples in Table 1. In contrast, when constructing VerifyBench-Hard, we deliberately enforced domain balance to ensure that **each domain contains approximately the same number of examples**.
>
> We also highlight that our benchmark construction methodology is highly generalizable. With access to large-scale datasets across more domains, our approach can readily generate additional evaluation samples to further expand and diversify VerifyBench. We hope to see more open-source datasets emerging from a wider range of domains in the future.

---

> > ### Author Response · Authors · 2025-11-23
> > **Rebuttal [2/2]**
> >
> > > **[Question 1]: RL training with Qwen-4B as the verifier clearly outperforms training with weaker models, while OSS does not yield additional gains. Does this suggest that there is in fact some “threshold” that determines whether a verifier is basically reasonable, and that once this threshold is reached, further gains become minor?**
> >
> > Thank you for your question. In Figure 3, the reason that using gpt-oss-20b as the verifier does not significantly outperform Qwen3-4B is that **the performance gap between these two models on VerifyBench is relatively small (2.8%)**. In contrast, the large gap between Qwen3-4B and other weaker models arises because their performance on VerifyBench differs substantially (10%–30%). In addition, we would like to point out that, as shown in Figure 3, gpt-oss-20b as a verifier slightly surpasses Qwen3-4B on MATH500 and MinervaMath, which is consistent with their smaller performance gap on VerifyBench.
> >
> > In fact, we believe it is difficult to determine a threshold that decides whether a verifier is effective, because **this depends heavily on the data used for RL training.** For example, if the training only uses queries whose answers are integers [1], the verification process becomes very simple. However, once we expand to a broader range of questions, we need to rely on model-based verifiers [2-3], and thus the verifier’s accuracy must be taken into account. **VerifyBench provides a direct and convenient way to evaluate model-based verifiers**, enabling researchers to improve the verification accuracy of their own verifiers.
> >
> > Ref:
> > [1] Yu et al. DAPO: An Open-Source LLM Reinforcement Learning System at Scale.
> > [2] Qwen Team, Qwen3 Technical Report.
> > [3] Meituan LongCat Team, Introducing LongCat-Flash-Thinking: A Technical Report.
> >
> > > **[Question 2]: In the RFT experiments, were models stronger than Qwen-4B (e.g., OSS used in the RL experiments) also evaluated? If so, how did they perform?**
> >
> > Thank you for your question. We have added RFT experiments based on gpt-oss-20b and updated Figure 10 accordingly in the new revision. As we mentioned in Question 1, since gpt-oss-20b and Qwen3-4B achieve comparable performance on VerifyBench (a difference of only 2.8%), using gpt-oss-20b for rejection sampling yields only marginal improvements over Qwen3-4B.
> >
> >
> > > **[Question 3]: Can you provide some challenging examples from VerifyBench and illustrate how the models judged them correctly or incorrectly? In particular, for the OSS model, examining its reasoning traces may clarify why it makes incorrect decisions. This could in turn inform future verifier improvements.**
> >
> > In the new revision, we have added more examples from VerifyBench-Hard (Appendix K), as well as cases where gpt-oss-120b makes verification errors on VerifyBench-Hard, including analyses of the model’s outputs and verification results (Appendix L).
> >
> > We sincerely appreciate you for pointing out this issue. We believe that adding these examples will help researchers better understand VerifyBench(-Hard), gain insights into why existing verifiers fail, and further improve the verification process.
> >
> > ---
> >
> > Thank you again for the thorough review. We think that our rebuttal can address your main concern, if you have any further questions, we would be happy to answer them.
> >
> > Sincerely,
> > VerifyBench Authors

---

### Official Review · Reviewer_8czc · 2025-10-25

**Soundness:** 3
**Presentation:** 3
**Contribution:** 3
**Rating:** 6
**Confidence:** 3

**Summary:**

This paper presents VerifyBench, a benchmark designed to evaluate the accuracy of reference-based reward systems. Specifically, given a question, reference answer and a generated completion, the benchmark tests if models are able to correctly classify the generated completion as correct/incorrect. A well-designed pipeline consisting of human annotation and large-scale model generations is used to curate VerifyBench and VerifyBench-Hard, a harder variant of the same task. Evaluations on the benchmark show that current models (especially smaller ones), have significant room for improvement on the task.

**Strengths:**

- The paper is well-written, smooth, and easy to follow, with clear structure and presentation.

- The data curation pipelines are comprehensive and well-designed.

- The benchmark tackles a challenge that is not addressed in existing benchmarks.

**Weaknesses:**

- The final source distribution of completions may be biased toward specific models, which could undermine benchmark diversity. More analysis on this would address this.

- Some more clarifications on the data curation pipeline will further increase clarity of the paper.

**Questions:**

1. What is the final source (model) distribution of completions? Models have a bias to prefer their own completions, so all completions coming from a small number of models will be undesirable.

2. Do completions include long reasoning traces (chains of thought) or only the final summarized solutions? Reasoning models generate long thinking traces involving complex behavior such as self-reflection/verification and backtracking. I personally do not think these traces should be part of the completion, instead the completion should be a final summary of the solution (Eg. Reasoning traces of GPT5 are not visible, only the final solution is presented to the user). I am curious to hear how authors approached this.

3. What does the length distribution of the completions in the benchmark look like?

4.  I liked the experiment of training with the best verifiers on the benchmark. Another experiment, which is much easier to run, is to use verifiers for rejection sampling (best-of-N). This can be compared to pass@N (using a perfect verifier). The difference in performance between the two methods would be another indicator of the strength of a verifier (an ideal verifier should closely match the pass@N curve). I believe this is also a cleaner experiment, as training might have second-order effects.

5. My main doubt is on the performance of reference answers. If reference answers are provided to the model, then isn’t the task simply just to extract the final answer from the completion and check equivalence to the reference answer. In domains like Math, this is exactly what a symbolic parser (e.g., Math-verify) does. This might be a harder task in non-math domains, so I am curious to understand the benefits of this in tasks like Math. In general, if one of the applications of these verifiers is RL training in domains where ground truth answers are unavailable, then a more practical evaluation is one without reference answers. Similarly, rolling out long traces from verifiers is also not practical for RL training, especially if slightly worse but significantly cheaper symbolic parsers are available.

6. Adding on to the previous point, if verifiers are to be used for inference scaling (best-of-N), then reference answers will not be available. Perhaps, the best thing to do is to expand the benchmark to have a reference and non-reference setting? This would be able to get the best of both worlds.

7. How exactly is controlled downsampling performed, and does it distort the difficulty/accuracy distribution of the dataset?

---

> ### Author Response · Authors · 2025-11-23
> **Rebuttal [1/4]**
>
> Dear Reviewer 8czc,
>
> Thank you for the effort and time you dedicated to reviewing our paper, and for your recognition of the contribution of our proposed methods and benchmarks. We have greatly benefited from your valuable feedback. Below are our responses to the questions you pointed out:
>
> ---
>
> > **[Question 1]: What is the final source (model) distribution of completions? Models have a bias to prefer their own completions, so all completions coming from a small number of models will be undesirable.**
>
> Thank you for your question. In Appendix F.1, we added the source model distribution for VerifyBench and VerifyBench-Hard. On VerifyBench, **the distribution of completions produced by different models does not exhibit extreme imbalance (each model contributes roughly 2–7%).**
>
> In addition, we included two evaluations to assess whether models exhibit a preference for their own completions. Specifically, for the two open-source models that are used both for generation and evaluation, Qwen2.5-72B-Instruct and Llama-3.3-70B-Instruct, we report their accuracy on completions generated by themselves versus completions generated by other models. The results are shown in the table below.
>
>
>
> | Model | VerifyBench (all) | VerifyBench (its completions) | VerifyBench (not its completions) | VerifyBench-Hard (all) | VerifyBench-Hard (its completions) | VerifyBench-Hard (not its completions) |
> | -------- | -------- | -------- |-------- |-------- |-------- |-------- |
> | Qwen2.5-72B-Instruct     | 92.05     | 93.02     | 92.01 | 62.20 | 50.00 | 62.42|
> | Llama-3.3-70B-Instruct     |  91.35     | 86.90     | 91.54 | / | / | /|
>
> From the limited results, the model does not exhibit a significant preference for the completions generated by itself. We believe that a reference-answer–based verification process is only weakly affected by such biased preferences. Moreover, **given the relatively balanced distribution of source models in our benchmark, we consider the bias introduced by this issue to be negligible.** Once again, we thank you for raising this point, which has helped us further improve our paper.
>
> > **[Question 2]: Do completions include long reasoning traces (chains of thought) or only the final summarized solutions? Reasoning models generate long thinking traces involving complex behavior such as self-reflection/verification and backtracking. I personally do not think these traces should be part of the completion, instead the completion should be a final summary of the solution (Eg. Reasoning traces of GPT5 are not visible, only the final solution is presented to the user). I am curious to hear how authors approached this.**
>
> Thank you for your question. We indeed considered this issue when constructing VerifyBench. Our final design choice was to **include the model-generated long reasoning traces in the completions. More precisely, we preserve the model’s raw output trajectories in their entirety.** Below, we explain the rationale behind this decision.
>
> In practical RL training, when assigning rewards, what we typically obtain are the raw reasoning rollouts generated by the model. At this stage, various post-processing methods may be applied to these raw trajectories, for example, removing the reasoning content inside `<think></think>`, or extracting the final answer enclosed in `\boxed{}`. We regard these operations as components of a broader "Reward System". For this reason, we title our work "Benchmarking Reference-based Reward Systems" rather than "Benchmarking Reference-based Reward Models". **Our goal is to evaluate verification methods that are aligned with real-world RL workflows, covering a wider spectrum of reward system components rather than only the reward model itself.**
>
> Concretely, for users of VerifyBench and VerifyBench-Hard, **we expect them to customize their own preprocessing pipelines, such as truncation, extraction, or parsing of completions.** Of course, we also provide a default preprocessing method in our scripts. Ultimately, the benchmark evaluates the accuracy of the entire reward system, including preprocessing steps, models, or regex-based functions, when performing verification.

---

> ### Author Response · Authors · 2025-11-23
> **Rebuttal [2/4]**
>
> > **[Question 3]: What does the length distribution of the completions in the benchmark look like?**
>
> We added statistics on the token-length distribution of the completions in Appendix F.2, and we briefly describe them here.
>
> |  | 0k-0.5k | 0.5k-1k | 1k-1.5k | 1.5k-2k | 2k-2.5k | 2.5k-3k | 3k-3.5k | 3.5k-4k | 4k+ |
> | - | - | - |- |- |- |- |- |- |- |
> | VerifyBench     | 909(45.45%) | 449(22.45%) | 199(9.95%) | 92(4.6%) | 61(3.05%) |28(1.4%) |32(1.6%) |36(1.8%) | 194(9.7%) |
> | VerifyBench-Hard     | 537(53.7%) | 197(19.7%) | 81(8.1%) | 47(4.7%) | 20(2.0%) |23(2.3%) |7(0.7%) |10(1.0%) | 78(7.8%) |
>
> It can be seen that the token-length range of the completions in our constructed VerifyBench is broad: it includes shorter chat-style completions as well as longer R1-like completions. **This diversity improves the coverage and realism of VerifyBench, making it a more reliable benchmark.** We sincerely appreciate your question, which helped us further refine our paper.
>
>
> > **[Question 4]: I liked the experiment of training with the best verifiers on the benchmark. Another experiment, which is much easier to run, is to use verifiers for rejection sampling (best-of-N). This can be compared to pass@N (using a perfect verifier). The difference in performance between the two methods would be another indicator of the strength of a verifier (an ideal verifier should closely match the pass@N curve). I believe this is also a cleaner experiment, as training might have second-order effects.**
>
> Thank you for your suggestion. We also considered performing an inference-time correlation analysis, similar to the Best-of-N (BoN) experiment conducted in RM-Bench [1]. However, VerifyBench evaluates *verification* tasks, which inherently require a reference answer. This characteristic makes it difficult to implement a BoN-style experiment. **If we omit the reference answer, the setup would deviate from the intended design of VerifyBench; if we include the reference answer, we would effectively leak the golden answer from the test set into the inference process, thereby invalidating the evaluation metrics on the test set.**
>
> Nonetheless, inspired by the idea of Rejection Sampling, we conducted a Rejection Sampling Fine-Tuning (RFT) experiment. Specifically, we perform large-scale rejection sampling on the training set and use only the accepted responses for SFT, then evaluate the fine-tuned model’s accuracy on downstream tasks. This experiment is presented in Appendix J.2. **The results show that using a verifier with stronger VerifyBench performance leads to better downstream performance after RFT, demonstrating the practical utility of VerifyBench.**
>
> Ref:
> [1] Liu, et al. RM-Bench: Benchmarking Reward Models of Language Models with Subtlety and Style.

---

> ### Author Response · Authors · 2025-11-23
> **Rebuttal [3/4]**
>
> > **[Question 5]: My main doubt is on the performance of reference answers. If reference answers are provided to the model, then isn’t the task simply just to extract the final answer from the completion and check equivalence to the reference answer. In domains like Math, this is exactly what a symbolic parser (e.g., Math-verify) does. This might be a harder task in non-math domains, so I am curious to understand the benefits of this in tasks like Math. In general, if one of the applications of these verifiers is RL training in domains where ground truth answers are unavailable, then a more practical evaluation is one without reference answers. Similarly, rolling out long traces from verifiers is also not practical for RL training, especially if slightly worse but significantly cheaper symbolic parsers are available.**
>
> Thank you for your question. We address it from several perspectives:
>
> + **On reference-based verification tasks:** We think that, in reasoning tasks, constructing accurate rule-based verifiers for all mathematical problems is inherently challenging. For example, if the golden truth is “John is 3 years older than Mary,” but the completion states “Mary is 3 years older than John,” Math-Verify will extract the number 3 from both the golden truth and the completion and incorrectly deem them equal, ignoring the semantic contradiction. Consequently, some prior works construct RL datasets that are easy to verify, such as those with integer answers [1]. However, **building more capable reasoning models requires handling more complex, diverse, and flexible reasoning tasks.** Current flagship models have already adopted LLM-based reference-based verifiers to compute rewards and deliver more precise feedback [2–4]. Furthermore, our overall evaluation results in Table 2 include Math-Verify, showing that even in its strongest categories, numeric values and expressions, its performance does not surpass state-of-the-art LLMs, supporting the claim above.
> + **On the computational cost of LLM-based verifiers:** Using LLMs for verification undeniably introduces additional computational overhead, especially when current LLM-verifiers also perform reasoning [3–4]. Nevertheless, we believe this is necessary for large-scale model training. Moreover, several optimization strategies exist: (1) we can iteratively improve LLM-verifiers to achieve strong verification accuracy with smaller models and fewer output tokens; (2) engineering optimizations can provide efficient, on-demand verification environments[5]; (3) hybrid reward systems combining rule-based verifiers with LLM-verifiers can be designed to invoke LLM-verification only when needed, enabling more efficient resource usage.
>
>
> Ref:
> [1] Yu et al. DAPO: An Open-Source LLM Reinforcement Learning System at Scale.
> [2] Qwen Team, Qwen3 Technical Report.
> [3] Meituan LongCat Team, Introducing LongCat-Flash-Thinking: A Technical Report.
> [4] ByteDance Seed, Seed1.5-Thinking: Advancing Superb Reasoning Models with Reinforcement Learning.
> [5] Ling Team, Every Step Evolves: Scaling Reinforcement Learning for Trillion-Scale Thinking Model.

---

> > ### Author Response · Authors · 2025-11-23
> > **Rebuttal [4/4]**
> >
> > > **[Question 6]: Adding on to the previous point, if verifiers are to be used for inference scaling (best-of-N), then reference answers will not be available. Perhaps, the best thing to do is to expand the benchmark to have a reference and non-reference setting? This would be able to get the best of both worlds.**
> >
> > Thank you for your question. In our original submission, we had already included an evaluation setting without reference answers. The corresponding results are reported in Section 4.2. We found that **removing the reference answer leads to a 5–18% accuracy drop for LLM-verifiers on VerifyBench, highlighting the critical role of reference answers in verification.**
> >
> > In addition, since VerifyBench is designed such that each query is paired with one correct completion and one incorrect completion, we can transform VerifyBench into a RewardBench-style [1] pairwise benchmark. We provide the corresponding experimental results in Section 4.3, which show that VerifyBench is substantially more challenging than existing benchmarks such as RewardBench [1] and RMBench [2].
> >
> > In summary, **the current version of VerifyBench already supports both reference and non-reference settings, enabling the research community to conduct evaluation from multiple perspectives.**
> >
> > Ref:
> > [1] Lambert, et al. RewardBench: Evaluating Reward Models for Language Modeling.
> > [2] Liu, et al. RM-Bench: Benchmarking Reward Models of Language Models with Subtlety and Style.
> >
> > > **[Question 7]: How exactly is controlled downsampling performed, and does it distort the difficulty/accuracy distribution of the dataset?**
> >
> > Thank you for your question. The number of candidate samples we constructed exceeded the planned benchmark size. The controlled downsampling described in Section 3.1 was primarily conducted to maintain two forms of balance:
> >
> > + **Answer type balance:** We designed four different answer types, each posing different levels of difficulty for verification. Accordingly, we **maintain an equal number of samples across these four answer types in VerifyBench**, enabling us to assess a model’s verification accuracy under different scenarios.
> >
> > + **Correctness balance:** Since verification is a binary classification task, we aim to minimize bias arising from class imbalance. **For each question in VerifyBench, we therefore include one correct completion and one incorrect completion.**
> >
> > This controlled downsampling may distort the original difficulty or accuracy distribution of the candidate samples, we believe it is essential for constructing a reasonable and effective benchmark.
> >
> >
> > ---
> >
> > Thank you again for the thorough review. We think that our rebuttal can address your main concern, if you have any further questions, we would be happy to answer them.
> >
> > Sincerely,
> > VerifyBench Authors

---

> > > ### Comment · Reviewer_8czc · 2025-11-24
> > >
> > > Thank you for the detailed rebuttal! My main concerns have been addressed, this benchmark is a timely and relevant addition. I have raised my score.

---

> > > > ### Author Response · Authors · 2025-11-25
> > > >
> > > > Thank you once again for your insightful comments and feedback! We will thoroughly revise and polish our paper, and we hope to have more discussions with you in the future. 😄

---

### Official Review · Reviewer_R9TK · 2025-10-30

**Soundness:** 3
**Presentation:** 3
**Contribution:** 2
**Rating:** 2
**Confidence:** 4

**Summary:**

This paper presents VerifyBench and a more challenging version VerfiyBench-Hard to assess reference-based reward systems used in training large reasoning models (LRMs). The authors develop a step-by-step pipeline to create these benchmarks. The authors evaluate a large pool of large language models (LLMs) and reward models (RMs) on these benchmarks.

**Strengths:**

1. The paper is well-written and easy to follow.
2. The proposed benchmarks, VerifyBench and VerifyBench-Hard, are well-designed to evaluate the capabilities of reward models in assessing the correctness of reasoning steps in LLM-generated outputs. These benchmarks could be valuable resources for the research community.

**Weaknesses:**

1. The motivation of the paper is somewhat questionable. Typically, the correctness of responses can be directly verified against ground truth answers without the need for reward models. Using a reward model to verify correctness can achieve higher accuracy compared to directly comparing with ground truth answers, as demonstrated in Figure 12, 13, and 14. However, it is unclear whether the added complexity of using reward models is justified given the performance gain.
2. The benchmark appears to be relatively easy for current state-of-the-art LLMs and RMs. As shown in Table 2, the best-performing LLMs and RMs achieve over 90% accuracy on VerifyBench, leaving limited room for improvement. In the evaluation, The verifier models are esstentially tasked to compare the consistency between the reference answer and the model-generated answer, which should be relatively straightforward for current LLMs.
3. In Section 5.2 and Appendix H, the authors demonstrate that better models on VerifyBench is a better verifier in RL training. However, it is unclear whether these models are better than directly matching with ground truth answers under the same computation budget.

**Questions:**

1. The authors demonstrate that better verifier models lead to better LLMs when used in RL training. What is the relationship between the verifier model's performance and the resulting LLM's performance? I am wondering whether weaker/smaller LLMs typically require stronger verifiers while stronger/larger LLMs can work well with weaker verifiers, such as direct matching with ground truth answers. Is this true?

---

> ### Author Response · Authors · 2025-11-23
> **Rebuttal [1/2]**
>
> Dear Reviewer R9TK,
>
> Thank you for the effort and time you dedicated to reviewing our paper, and for your recognition of the contribution of our proposed benchmarks. We have benefited from your valuable feedback. Below are our responses to the weaknesses you pointed out:
>
> ---
>
> > **[Weakness 1]: The motivation of the paper is somewhat questionable. Typically, the correctness of responses can be directly verified against ground truth answers without the need for reward models. Using a reward model to verify correctness can achieve higher accuracy compared to directly comparing with ground truth answers, as demonstrated in Figure 12, 13, and 14. However, it is unclear whether the added complexity of using reward models is justified given the performance gain.**
>
> Thank you for your question. We think that, in reasoning tasks, constructing accurate rule-based verifiers for all mathematical problems is inherently challenging. For example, if the golden truth is "John is 3 years older than Mary", but the completion states "Mary is 3 years older than John", Math-Verify will extract the number 3 from both the golden truth and the completion and incorrectly deem them equal, ignoring the semantic contradiction.
>
> Consequently, some prior works construct RL datasets that are easy to verify, such as those with integer answers [1]. However, **building more capable reasoning models requires handling more complex, diverse, and flexible reasoning tasks. Current flagship models have already adopted LLM-based reference-based verifiers to compute rewards and deliver more precise feedback [2–4]**.
>
> Furthermore, our overall evaluation results in Table 2 include Math-Verify, a widely-used rule-based verification method, showing that even in its strongest categories, numeric values and expressions, its performance does not surpass state-of-the-art LLMs, supporting the claim above.
>
> We provide hard-to-verify samples from VerifyBench-Hard in Appendix L, and present the reasoning traces of gpt-oss-120b, one of the best-performing open-source models on VerifyBench, in Appendix M. These additions offer a clearer understanding of the motivation behind VerifyBench and its practical value.
>
>
> Ref:
> [1] Yu, et al. DAPO: An Open-Source LLM Reinforcement Learning System at Scale.
> [2] Qwen Team, Qwen3 Technical Report.
> [3] Meituan LongCat Team, Introducing LongCat-Flash-Thinking: A Technical Report.
> [4] ByteDance Seed, Seed1.5-Thinking: Advancing Superb Reasoning Models with Reinforcement Learning.
>
>
> > **[Weakness 2]: The benchmark appears to be relatively easy for current state-of-the-art LLMs and RMs. As shown in Table 2, the best-performing LLMs and RMs achieve over 90% accuracy on VerifyBench, leaving limited room for improvement. In the evaluation, The verifier models are esstentially tasked to compare the consistency between the reference answer and the model-generated answer, which should be relatively straightforward for current LLMs.**
>
> Thank you for raising this question.
>
> First, we would like to emphasize that VerifyBench is a binary classification task. Due to our balanced correctness design, a random guess should achieve around 50% accuracy. **VerifyBench is not designed to "stump" LLMs; instead, it aims to help researchers comprehensively understand their verifiers under real scenarios**. Since the queries and completions in VerifyBench come from natural sampling, the benchmark provides an objective and holistic evaluation of a verification system’s performance under diverse conditions.
>
> Given that current best-performing LLMs and RMs achieve over 90% accuracy on VerifyBench, we constructed VerifyBench-Hard, which contains significantly more hard-to-verify tasks. It is built via difficulty-based sampling, and **most existing models cannot surpass 80% accuracy on this harder benchmark**.
>
> In addition, we want to highlight that reward systems must operate at scale during RL training. Using state-of-the-art LLMs or RMs as verifiers requires substantial computational resources and introduces considerable latency. Thus, in Section 4.1, **we point out that "Small-scale models still have room for development."** Current small-scale models (<3B) achieve only ~80% accuracy even on the easier VerifyBench. We believe that future research should focus **not only on improving verification accuracy, but also on achieving high-accuracy verification with significantly lower computational cost**.
>
> Regarding the difficulty of the verify task, this concern is similar to Weakness 1, and we have provided a detailed response in our reply to Weakness 1.

---

> > ### Author Response · Authors · 2025-11-23
> > **Rebuttal [2/2]**
> >
> > > **[Weakness 3]: In Section 5.2 and Appendix H, the authors demonstrate that better models on VerifyBench is a better verifier in RL training. However, it is unclear whether these models are better than directly matching with ground truth answers under the same computation budget.**
> >
> > Thank you for pointing this out. During the rebuttal period, we added experimental results that use **math-verify**, a widely adopted rule-based function, as the verifier in Section 5.2 and Appendix H. The corresponding figures in the revision have been updated accordingly.
> >
> > From the results, we observe that **using gpt-oss-20b as the verifier leads to consistently better downstream performance in both the RL and RFT experiments**.
> >
> > We would also like to highlight that the datasets used in our RL and RFT experiments consist of relatively easy-to-verify mathematical problems, for which math-verify has been specifically optimized. However, **for more diverse and general RL datasets, the accuracy of math-verify drops substantially (as shown in Table 2).**
> >
> > > **[Question 1]: The authors demonstrate that better verifier models lead to better LLMs when used in RL training. What is the relationship between the verifier model's performance and the resulting LLM's performance? I am wondering whether weaker/smaller LLMs typically require stronger verifiers while stronger/larger LLMs can work well with weaker verifiers, such as direct matching with ground truth answers. Is this true?**
> >
> > Thank you for your question.
> >
> > We believe that a model’s capability is not correlated with whether it requires a stronger or weaker verifier. Incorrect rewards can steer the model’s outputs toward unexpected directions, and for a powerful actor LLM, providing high-noise reward signals can still undermine the effectiveness of RL. Therefore, we observe that current flagship models also invest in developing stronger verifiers and rely on them for RL training. For example, LongCat-Flash-Thinking contains ~500B parameters, and Seed1.5-Thinking contains ~200B parameters; both leverage LLM-based verifiers. This indicates that regardless of model size, stronger verifiers are essential for RL training.
> >
> > To further support our claim, we conducted RL experiments on Qwen2.5-7B-Instruct, a larger and more capable model. Specifically, we used math-verify, Qwen3-1.7B, and gpt-oss-20b as verifiers under the same RL setup as Section 5.2. We found that when using Qwen2.5-7B-Instruct as the base model, the observed trends are consistent with those reported for Qwen2.5-3B-Instruct (used in the paper). The corresponding results are shown in the table below.
> >
> > | Base model | verifier | MATH500 | AMC23 | MinervaMath |
> > | -------- | -------- | -------- | -------- | -------- |
> > | Qwen2.5-3B-Instruct     | math-verify (Rule-based)     | 65.43 -> 67.97     | 39.69 -> 43.43 | 27.76 -> 28.48 |
> > | | Qwen3-1.7B | 65.43 -> 60.43 | 39.69 -> 37.50 | 27.76 -> 24.08|
> > | | gpt-oss-20b| 65.43 -> 68.67 | 39.69 -> 45.94 | 27.76 -> 29.31|
> > | Qwen2.5-7B-Instruct     | math-verify (Rule-based)     | 75.98 -> 78.32     | 54.06 -> 59.14 | 35.80 -> 38.74 |
> > | | Qwen3-1.7B | 75.98 -> 72.17 | 54.06 -> 52.28 | 35.80 -> 32.55|
> > | | gpt-oss-20b| 75.98 -> 79.02 | 54.06 -> 59.19 | 35.80 -> 39.04|
> >
> > These results show that RL fails to perform well on a stronger base model when paired with weaker verifiers, thereby providing additional evidence for our conclusion.
> >
> >
> > ---
> >
> > Thank you again for the thorough review. We think that our rebuttal can address your main concern, if you have any further questions, we would be happy to answer them.
> >
> > Sincerely,
> > VerifyBench Authors

---

> ### Comment · Reviewer_R9TK · 2025-11-28
>
> Thank you for your detailed response.
>
> In terms of the computational budget, using RM introduces extra cost, while the cost of using math-verify, which is basically string matching, is almost negligible. Suppose the cost on LLM is X and the cost on RM is Y, what if we train the LLM with math-verify with the cost of X+Y? Correct me if I miss something in the submission.
>
> Furthermore, I am still concern about the difficulty of this benchmark, as ~80% accuracy still appears to be very high to me and this benchmark is very likely to be saturated soon.
>
> Generally, given the current response, I would like to raise my score from 2 to 4. Please note that the EDIT button appears to be removed by the OpenReview system. Please remind me when it is added back.

---

> ### Author Response · Authors · 2025-12-01
>
> Thank you for your reply and your kind re-consideration of the score. We would like to further clarify the two concerns you mentioned.
>
> ---
>
> > **Q1: Regarding computational budget—using an RM introduces additional cost, whereas math-verify (essentially string matching) incurs almost negligible cost.**
>
> We would like to respond to this question from three perspectives:
>
> + **Evaluation goal of VerifyBench:** The purpose of VerifyBench is to evaluate reference-based reward systems. These systems are not limited to LLM-based verifiers, they may also be rule-based or combined designs. Our goal is to assess the accuracy of such systems on verification tasks; computational efficiency is currently outside the scope of this benchmark.
>
> + **Model-based verifiers are widely used in RL training:** As discussed in our rebuttal, state-of-the-art models increasingly rely on reference-based LLM verifiers to obtain more robust and accurate validation signals. This has already become a common practice when building more general reasoning models.
>
> + **Cost of model-based verifiers:** We believe that the additional computation required by model-based verifiers during RL training is marginal. Compared with the substantial decoding cost of the actor model, the overhead introduced by a small LLM-verifier, especially after applying carefully designed preprocessing of the completions, is minimal in terms of latency.
>
>
> > **Q2: Regarding the difficulty of the benchmark**
>
> As outlined in our rebuttal, we would like to further clarify this point from two additional angles:
>
> + **Difficulty of VerifyBench:** We constructed a high-difficulty split (VerifyBench-Hard), on which the accuracy of most models remains below 80%. We believe there is still significant room for improvement, and saturation is unlikely in the near term.
>
> + **Substantial room for improvement for small models:** Because LLM verifiers are used in real time during RL training, model size directly affects training efficiency. Therefore, researchers should pay more attention to developing small LLM verifiers. Even on the “easy” version of VerifyBench, current small models (<3B parameters) still have large headroom for progress, with accuracies remaining below 70%.
>
> ---
>
> Once again, we sincerely thank you for your reply and for the time and effort you have invested during the review process. We look forward to more opportunities for discussion in the future.
>
> Thank you!

---

### Author Response · Authors · 2025-12-01
**Summary of Rebuttal & Discussion phases**

Dear Reviewers, AC, SAC, and PC,

We sincerely appreciate the time and effort you devoted to reviewing our work, as well as the insightful feedback you provided, which has significantly strengthened our paper. Below, we summarize the progress so far.

---
We are encouraged to see that the reviewers recognized several key strengths of our work, including:

+ **Well-motivated benchmarks:** Our proposed VerifyBench and VerifyBench-Hard, are well-motivated to evaluate the capabilities of reward models in assessing the correctness of LLM-generated outputs. (Reviewer `R9TK`, `ApHj`)
+ **Well-designed pipeline for benchmark construction:** The well-designed pipeline ensures high diversity, and the use of human annotators for verification improves reliability and validity. (Reviewer `8czc`, `uUti`)
+ **Timely contribution:** Our work fill a clear gap in the evaluation landscape, as the community shifts from preference-based to reference-based reward in RL for reasoning models. (Reviewer `8czc`, `ApHj`, `uUti`)
+ **Strong correlation with RL performance:** The RL correlation experiment convincingly emphasize the benchmark’s practical utility.(Reviewer `ApHj`, `uUti`)
+ **Well-written and easy to understand:** The writing and presentation are clear, making the paper easy to follow. (Reviewer `R9TK`, `8czc`)

---
During the rebuttal phase, we conducted additional experiments or supplemented relevant analyses in the revision to address the reviewers' concerns:

+ We further incorporate correlation studies using a diverse set of rule-based and model-based verifiers, covering both RFT and real RL experiments. The results demonstrate that **verifiers achieving stronger performance on VerifyBench consistently translate into improved downstream task performance**. (Reviewer `R9TK`, `ApHj`)
+ We additionally analyze models’ preferences toward their own completions. Owing to the presence of reference answers in VerifyBench, we show that **the generating model exhibits only a negligible preference for its own completions during evaluation.** (Reviewer `8czc`)
+ We added more illustrative evaluation examples from VerifyBench and VerifyBench-Hard in Appendices K and L to help readers better understand and familiarize themselves with our proposed benchmarks. (Reviewer `R9TK`, `ApHj`)
+ In Appendix M, we included an error case study of gpt-oss-20b on VerifyBench, which **demonstrates why the model fails during verification and provides insights for future improvements in model verification capabilities.** (Reviewer `R9TK`)
+ We supplemented Appendix F with additional benchmark statistics, including the distribution of completion sources and the distribution of token counts for the evaluation samples. (Reviewer `8czc`)

---
We also clarified several key points to address the reviewers' concerns:

+ We explained the importance of model-based verifiers, including providing realistic examples and the RL training recipe of existing SOTA models, in order to demonstrate the practical utility of VerifyBench. (Reviewer `R9TK`, `8czc`)
+ We conducted further analysis and explanation of the empirical observations from the RL-correlation experiments, illustrating that evaluation through verifiers on VerifyBench is predictive of actual RL performance. (Reviewer `ApHj`, `uUti`)
+ We provided additional analysis and interpretation of existing evaluation results on VerifyBench and VerifyBench-Hard, including discussions of the difficulty of VerifyBench, distinctions among different categories of verifiers, and insights revealed by the benchmark results. (Reviewer `R9TK`, `uUti`)
+ We introduced the domains currently covered in VerifyBench. At present, VerifyBench spans math, logic, and general reasoning, ensuring both coverage and diversity with respect to models’ reasoning capabilities. (Reviewer `ApHj`, `uUti`)
+ We provide a more detailed explanation of the benchmark construction process and the evaluation settings. (Reviewer `8czc`, `uUti`)

---
We would like to briefly summarize the consensus reached with the reviewers during the discussion phase. After the discussion:
+ `Reviewer uUti` stated that *this is a good work with interest for the community* and maintained their rating of **8**.
+ `Reviewer 8czc` felt that our detailed rebuttal satisfactorily resolved their main concerns and increased their rating from **6** to **8**.
+ `Reviewer R9TK` considered some of their concerns to be addressed and was willing to raise their score from **2** to **4**, but due to current ICLR policy, we were unable to engage in further discussion.

In summary, before all the reviews reverted, our ratings improved from **2/6/6/8** to **4/6/8/8**.

---
Once again, we sincerely thank all reviewers for their recognition and valuable suggestions. We believe that our responses address the raised concerns effectively, and we hope the program committee (reviewers, ACs, SACs, PCs) will take them into account in the next phases.

Sincerely,
VerifyBench Authors

---

### Meta-Review · Area_Chair_xzdb · 2026-01-02

**Summary:**

This work introduces two benchmarks, VerifyBench and VerifyBench-Hard, to assess the performance of reference-based reward systems. Additionally, the authors conduct a thorough and comprehensive analysis of evaluation results, offering insights for understanding and developing reference-based reward systems. The proposed benchmarks can potentially serve as effective tools for guiding the development of verifier accuracy and the reasoning capabilities of models trained via RL in reasoning tasks.

Most reviewers acknowledge that the proposed benchmark is a well-motivated and timely contribution. Reviewer R9TK raised several concerns, and the authors provided corresponding responses during the rebuttal phase. Overall, I am inclined to accept this paper. However, following Reviewer R9TK’s suggestions, the authors may consider further increasing the difficulty of the benchmark in future work.

**Reviewer Concerns:**

Most reviewers’ concerns have been addressed.

**Reviewer Scores:**

Reviewer R9TK has stated in the discussion that they will revise their score.

---

### Decision · Program_Chairs · 2026-01-26

Accept (Poster)